# scChord: A Probabilistic Manifold Rectification Framework for RNA-to-Protein Translation

**Jiawei Zhang** [1]  **Kangjun Jin** [1]  **Shuai Xiao** [1]  **Jiachen Yang** [1]

## Abstract

Measuring single-cell protein abundance is essential for resolving biological mechanisms and disease progression with high resolution. However, due to the high costs and antibody throughput limitations of current proteomics, inferring protein levels from readily available RNA data has become a critical computational necessity. Existing regression and generative methods face a fundamental geometric bottleneck: enforcing deterministic constraints on noisy, heteroscedastic data collapses intrinsic uncertainty into a rough latent manifold, which destabilizes the learning process. To overcome this, we present *scChord*, a noise-decoupled conditional flow matching framework built on Probabilistic Manifold Rectification. Our approach utilizes a probabilistic decoder to disentangle technical noise and over-dispersion from the raw counts, absorbing them into distributional parameters. This allows the rectified latent manifold to focus more on biological signals, serving as a robust geometric regularizer for learning efficient transport trajectories. Extensive experiments on multiple multi-omics benchmarks demonstrate that scChord not only achieves state-of-the-art inference accuracy but also faithfully reconstructs high-fidelity biological heterogeneity and complex protein distributions.

## 1. Introduction

The central dogma of molecular biology describes the flow of genetic information from DNA to RNA and finally to proteins, the functional effectors of cellular life (Schoof et al., 2021). While single-cell transcriptomics (scRNA-seq) has become ubiquitous for profiling gene expression, it offers

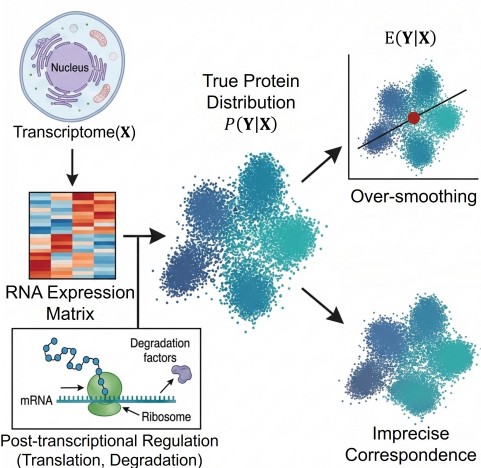

*Figure 1.* Challenges in inferring protein abundance. Post-transcriptional stochasticity induces a complex distribution $P(Y|X)$. Existing methods fail by either fitting the conditional mean or misaligning states under noise.

only a partial view of cellular phenotypes (Tang et al., 2009). Protein abundance often diverges significantly from transcript levels due to complex, non-linear post-transcriptional regulations, such as translation efficiency, degradation rates, and protein half-lives (Liu et al., 2016). Consequently, mapping the transcriptomic landscape to the proteomic state is not merely a data imputation task but a fundamental stochastic inverse problem essential for resolving functional heterogeneity (Vogel & Marcotte, 2012). Although multi-omics technologies like CITE-seq can simultaneously measure both modalities, their high cost and antibody throughput limitations necessitate computational methods to infer protein abundance from readily available RNA data (Stoeckius et al., 2017; Lakkis et al., 2022).

Current approaches to this translation task predominantly fall into two categories: deterministic regression and standard generative modeling (Hu et al., 2024). Deterministic methods learn point-wise mappings by minimizing reconstruction error. However, single-cell data is inherently heteroscedastic and over-dispersed, characterized by technical dropouts and biological stochasticity. Regression models inevitably learn the conditional mean, leading to over-

[1] School of Electrical and Information Engineering, Tianjin University, Tianjin, China. Correspondence to: Shuai Xiao <xs611@tju.edu.cn>, Kangjun Jin <jinkangjun@tju.edu.cn>.

smoothed predictions that miss multimodal protein distributions (Bishop, 1994; Lance et al., 2022). On the other hand, generative models like Variational Autoencoders (VAEs) aim to model the distribution but often struggle to establish precise, condition-specific correspondences (Bond-Taylor et al., 2021; Townes et al., 2019).

Recent advances in dynamical generative models (e.g., diffusion models (Ho et al., 2020) and flow matching (Lipman et al., 2022)) learn continuous stochastic/deterministic dynamics to represent complex distributions, showing strong potential in representation learning (Song et al., 2020). However, when applied to inherently uncertain biological translation tasks, these methods do not consistently outperform classical regression or probabilistic modeling baselines (Yu et al., 2024). We argue this stems from a mismatch between task geometry and modeling assumptions.

In particular, standard deterministic constraints that enforce rigid point-wise correspondences inevitably treat intrinsic biological uncertainty and technical noise as signal to be fitted (Kharchenko et al., 2014). Our analysis reveals that such constraints force the encoder to embed high-frequency noise into the latent space, effectively collapsing stochastic uncertainty into geometric irregularities on the latent manifold. On this rough landscape, the induced target vector field exhibits sharp fluctuations, resulting in high-curvature trajectories that render the learning of continuous dynamics numerically unstable and computationally burdensome (De Bortoli, 2022; Palma et al., 2025).

To overcome this barrier, we propose **scChord**, a two-stage framework for RNA-to-protein translation as conditional transport on a rectified latent manifold. Stage 1 (*Probabilistic Manifold Rectification*) uses a probabilistic decoder to absorb measurement uncertainty into distributional parameters, smoothing latent geometry; Stage 2 learns RNA-conditioned transport via consistency-regularized CFM on this rectified space.

Our contributions are threefold: (i) we characterize a geometric failure mode of deterministic constraints on stochastic single-cell measurements, linking uncertainty collapse to rough manifolds and stiff vector fields; (ii) we introduce Probabilistic Manifold Rectification to disentangle noise from latent geometry by decoupling uncertainty into distributional parameters; and (iii) we develop a consistency-regularized OT-CFM translator and show improved accuracy and distributional fidelity across multiple benchmarks.

**Conflict of Interest Disclosure** The authors declare that they have no known competing financial interests or personal relationships that could have appeared to influence the work reported in this paper.

## 2. Related work

Existing approaches range from classical *algorithmic alignment*, exemplified by Seurat's anchor-based matching (Hao et al., 2024) and LIGER's joint matrix factorization (Welch et al., 2019), to *learning-based* frameworks. The latter typically follow two paths: (i) deterministic regression and (ii) conventional conditional generative modeling.

Deterministic methods enforce point estimates, including cTP-net (Zhou et al., 2020), kernel/low-rank baselines (Lance et al., 2022), and neural translators such as BABEL (Wu et al., 2021) and sciPENN (Lakkis et al., 2022). Graph-based variants (Wen et al., 2022; Dai et al., 2021) reduce average error but still recover conditional means, under-representing heavy tails and rare cellular states. Recently, foundational models such as scTranslator (Liu et al., 2025) have emerged, leveraging large-scale pre-training to achieve robust cross-modal translation across diverse biological settings.

Conventional generative approaches instead model $p(y \mid x)$ using latent-variable decoders. totalVI (Gayoso et al., 2021) introduced probabilistic count likelihoods for CITE-seq to account for background noise and over-dispersion, and subsequent work extended this backbone to reference mapping and missing-modality settings (Du et al., 2022; Lotfollahi et al., 2022; Ashuach et al., 2023). These methods primarily maximize a static likelihood under simple gaussian priors via probabilistic reconstruction. In contrast, scChord uses a probabilistic decoder as a geometric rectifier that absorbs measurement noise and constructs a smooth latent target for downstream transport.

Modern generative dynamics have evolved from diffusion models (Ho et al., 2020) toward Flow Matching (Lipman et al., 2022) and OT-based Conditional Flow Matching (Tong et al., 2023; Liu et al., 2022), with the goal of learning continuous transport maps with efficient ODE trajectories. In biological applications, diffusion-based models have shown strong fidelity for generation and annotation in domains such as medical imaging (Kazerouni et al., 2023; Li et al., 2023); however, naively transferring these frameworks to conditional cross-modal translation like scDM (Yu et al., 2024) can be brittle under sparse and over-dispersed measurements, often degrading performance and increasing sampling cost. By coupling RNA-conditioned flow transport with probabilistic manifold rectification, scChord bridges these directions and enables stable dynamics together with controllable sampling for RNA-to-protein prediction.

## 3. scChord

Let $\mathcal{X} \subseteq \mathbb{R}^{d_{rna}}$ and $\mathcal{Y} \subseteq \mathbb{N}_0^{d_{prot}}$ denote the feature spaces of transcriptomic profiles and proteomic abundances, respectively. We identify a critical geometric obstruction in

applying continuous dynamics to discrete multi-omics data: the rough latent geometry induced by the collapse of intrinsic uncertainty. To overcome this, scChord employs a two-stage coupled framework: (i) **Probabilistic Manifold Rectification**, which leverages a variational objective to disentangle technical noise from the latent geometry, and (ii) **Consistency-Regularized Flow Matching**, which learns transport dynamics on this rectified smooth manifold.

### 3.1. The Geometry of Stochastic Heterogeneity

While Flow Matching theoretically enables learning arbitrary probability paths, its application to single-cell proteomics is obstructed by the dissonance between the deterministic nature of the learned vector field and the stochastic properties (Elowitz et al., 2002) of the data.

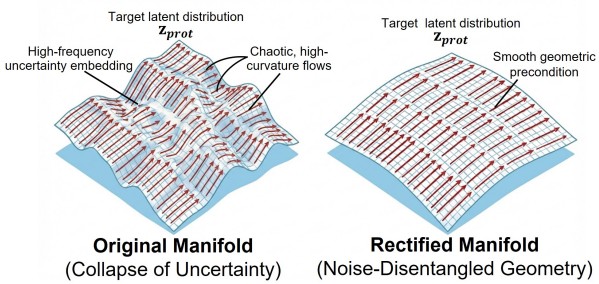

*Figure 2.* Probabilistic Manifold Rectification projects noisy, overdispersed protein counts onto a smooth latent manifold, enabling stable flow learning.

Single-cell protein counts are stochastic and heteroscedastic (Lähnemann et al., 2020).Assuming a continuous relaxation for the count space, We model paired data as

$$y = f(x) + \xi, \qquad \mathbb{E}[\xi \mid x] = 0, \ \text{Var}(\xi \mid x) = \Sigma(x), \ (1)$$

where $x$ is RNA, $f$ is the latent biological map, and $\xi$ aggregates post-transcriptional randomness and technical noise. Because $\text{tr}(\Sigma(x))$ is non-negligible, biologically similar cells can produce different protein realizations, forming a data manifold characterized by high local variance.

The key obstruction is that any deterministic encoder $g$ that preserves nontrivial variation across distinct realizations must become locally sensitive (Szegedy et al., 2013).

Formally, if for some $x$ two independent samples $y_1, y_2 \sim p(\cdot \mid x)$ satisfy $\|g(y_1) - g(y_2)\| \geq \delta$ with nontrivial probability (Sokolić et al., 2017), then the mean value inequality implies the existence of a point on the segment $[y_1, y_2]$ where the local sensitivity is lower-bounded by the ratio of output to input distances,

$$\|\nabla_y g(\bar{y})\|_{\text{op}} \ \geq \ \frac{\|g(y_1) - g(y_2)\|}{\|y_1 - y_2\|}. \qquad (2)$$

Appendix A.1 states this bound precisely and provides a proof sketch. In noisy regions, such sensitivity induces a highly irregular latent geometry, resulting in stiff vector fields and increased ODE solver costs (Chen et al., 2018), which we verify in Section 4.2.

### 3.2. Stage 1: Probabilistic Manifold Rectification

To construct a smooth manifold $\mathcal{M}_{\text{lat}}$, we must decouple the latent geometry from measurement noise. Rather than treating deviations as structural features, we postulate that technical noise can be statistically decoupled from the underlying biological state. We therefore employ a *Variance-Adaptive Gradient Rescaling* strategy via a ZINB decoder to separate these factors.

The decoder parameterizes the conditional likelihood $p_\psi(y|z)$ using a mean $\mu(z)$, a gene-wise dispersion $\theta$, and a dropout probability $\pi(z)$. The optimization objective is the Evidence Lower Bound (ELBO):

$$\mathcal{L} = -\mathbb{E}_{q_\phi(z|y)}[\log p_\psi(y|z)] + \beta D_{\text{KL}}(q_\phi(z|y)\|\mathcal{N}(0, I)). \tag{3}$$

Crucially, this probabilistic formulation alters the geometry of the optimization landscape compared to deterministic approaches. While deterministic modeling (e.g., via MSE loss) imposes an isotropic gradient field that forces the encoder to fit every observation regardless of its reliability, the ZINB likelihood introduces a dynamic mechanism that modulates the encoder's sensitivity based on data uncertainty.

We term this mechanism *Variance-Adaptive Gradient Rescaling*. As analytically derived in Appendix A.2, the magnitude of the gradient propagated to the latent variable $z$ is not constant but is scaled inversely proportional to the estimated uncertainty. Specifically, the effective gradient norm is bounded by the posterior signal confidence $\gamma(z)$ and a dispersion-dependent scaling factor $\lambda(\mu, \theta)$:

$$\|\nabla_z \mathcal{L}_{\text{ZINB}}\| \leq \gamma(z) \cdot \lambda(\mu, \theta) \cdot \|\nabla_z \mu\| \cdot \|y - \mu\|. \quad (4)$$

This inequality implies that in regimes of high heteroscedasticity (small $\theta$) or zero-inflation ($\pi \to 1$), the scaling factors $\gamma$ and $\lambda$ vanish, effectively detaching the encoder update from stochastic outliers. The term $\gamma(z)$ acts as a soft gate that suppresses gradients for likely dropout events, while $\lambda(\mu, \theta)$ dampens the signal from over-dispersed counts. Consequently, in noisy regions, the reconstruction gradient diminishes, allowing the geometric prior (KL divergence) to dominate. This implicitly minimizes the encoder's local Lipschitz constant, projecting the noisy data onto a rectified, smooth manifold suitable for stable flow matching.

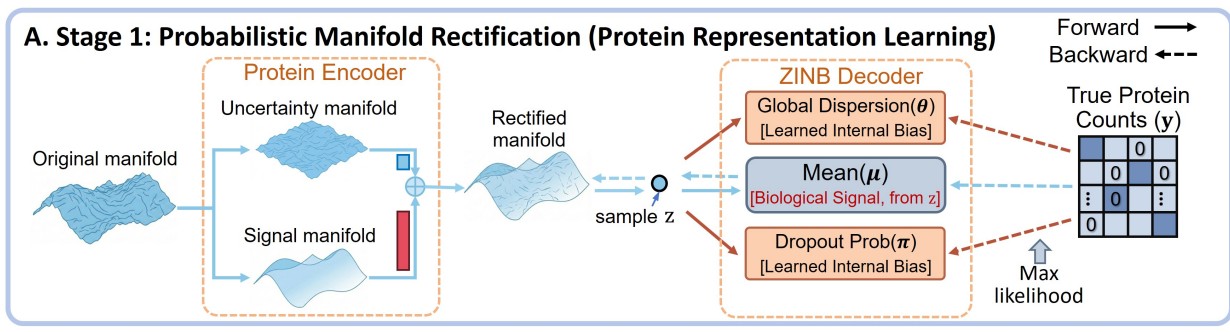

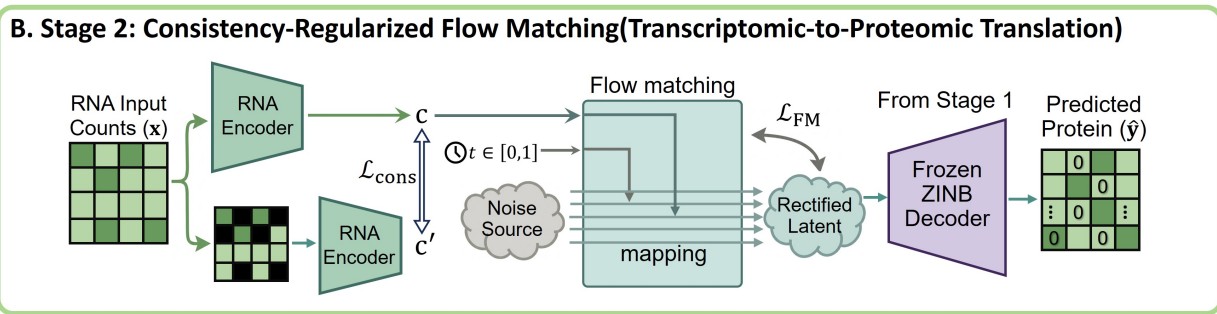

*Figure 3.* Overview of the scChord framework. **(A) Stage 1: Probabilistic Manifold Rectification** learns a smooth latent representation of protein counts by absorbing technical noise into a probabilistic decoder. **(B) Stage 2: Consistency-Regularized Flow Matching** learns an optimal transport map from RNA to protein on the rectified manifold, regularized for robustness to input sparsity.

### 3.3. Stage 2: Consistency-Regularized Flow Matching

Building upon the rectified manifold established in Stage 1, the second stage focuses on learning a conditional generative trajectory from noise to protein states. However, a critical challenge remains: the conditioning scRNA-seq inputs are inherently sparse and noisy. Therefore, our framework prioritizes a *transcriptomic consistency* mechanism to ensure robust conditioning before defining the generative flow dynamics.

The quality of conditional generation hinges on the stability of the input representation. Due to technical dropouts, biologically identical cells often exhibit vastly different RNA profiles. If the model conditions directly on these artifacts, the learned vector field becomes inconsistent.

To address this, we introduce a self-supervised consistency constraint. We simulate technical noise by randomly masking a ratio $r \in [r_{\min}, r_{\max}]$ of the expressed genes in the input $\mathbf{x}^{\text{rna}}$, yielding a corrupted view $\tilde{\mathbf{x}}^{\text{rna}}$. We then enforce the embedding of this corrupted view to align with that of the full profile:

$$\mathcal{L}_{\text{cons}} = \|\text{Enc}(\mathbf{x}^{\text{rna}}) - \text{Enc}(\tilde{\mathbf{x}}^{\text{rna}})\|_2^2. \quad (5)$$

By minimizing $\mathcal{L}_{\text{cons}}$, the encoder is forced to capture robust biological semantics rather than specific non-zero patterns. This ensures that the subsequent flow matching is guided by the underlying cellular state, enhancing inference stability on unseen, sparse data.

Equipped with these noise-invariant embeddings, we train a conditional Continuous Normalizing Flow (CNF) to transport latent samples from a standard Gaussian prior $x_0 \sim \mathcal{N}(0, I)$ to the rectified protein posterior $x_1$. Crucially, the target $x_1$ is sampled from the pre-trained Stage 1 VAE posterior $q_\phi(z|y)$, which provides a smoothed and high-density region of the latent space.

We employ the Optimal Transport Conditional Flow Matching (OT-CFM) objective (Tong et al., 2023). The transport trajectory is defined by the straight linear interpolation path $z_t = (1 - t)x_0 + tx_1$. The flow network $v_\theta$ is trained to regress the constant target vector field $u_t = x_1 - x_0$ conditioned on the robust RNA embedding $c$:

$$\mathcal{L}_{\text{CFM}} = \mathbb{E}_{t,x_0,x_1,c}\left[\|v_\theta(z_t, t, c) - (x_1 - x_0)\|^2\right]. \quad (6)$$

To further enhance generation quality and controllability, we incorporate Classifier-Free Guidance (CFG) during training by randomly dropping the condition $c$ with probability $p_{\text{uncond}}$. The final training objective combines the flow matching loss with the consistency regularization:

$$\mathcal{L}_{\text{total}} = \mathcal{L}_{\text{CFM}} + \lambda_{\text{cons}}\mathcal{L}_{\text{cons}}, \quad (7)$$

where $\lambda_{\text{cons}}$ balances the trade-off between trajectory learning and representation robustness.

### 3.4. Why Decoupled Two-Stage Training?

We decouple Stage 1 and Stage 2 instead of training them end-to-end because the two objectives serve different roles.

Stage 1 is designed to absorb measurement noise into a probabilistic decoder and yield a rectified protein manifold, whereas Stage 2 assumes that this manifold is already stabilized and then learns a flow on top of it. If these stages are jointly optimized from scratch, the CFM loss can push gradients directly into the protein encoder and partially undo the noise-absorbing effect of the ZINB decoder, so the encoder again has an incentive to encode measurement noise rather than suppress it.

A second failure mode is target drift. In joint training, the Stage 1 posterior mean continues to move while Stage 2 is trying to fit the flow field, so the endpoint $x_1$ is no longer stationary. This moving-target behavior makes the vector-field regression problem less well posed, increases trajectory curvature, and typically raises solver stiffness during sampling. In contrast, pretraining Stage 1 first provides a fixed rectified target manifold for CFM, which empirically leads to more stable optimization and better-conditioned flows.

These considerations also clarify the controlled comparison in Appendix **??**: freezing the pre-trained encoder degrades performance, while fine-tuning it during CFM remains competitive. Overall, the comparison suggests that the schedule is not a cosmetic implementation detail; it directly affects how well Stage 1 can absorb noise before Stage 2 learns the transport field.

## 4. Experiments

We systematically evaluate scChord on CITE-seq and ECCITE-seq datasets, covering prediction accuracy, distributional fidelity, robustness to batch effects and input sparsity, and scalability to large cohorts. We further conduct mechanism studies and ablation analyses to elucidate how *Probabilistic Manifold Rectification* improves latent geometry and enables stable flow-matching dynamics. Finally, we provide qualitative visualizations demonstrating scChord's high-fidelity reconstruction of complex protein expression distributions. Both stages utilize the same dataset splits to avoid data leakage. Full dataset descriptions, experimental settings, and metric definitions are deferred to Appendix B.

### 4.1. Performance Comparison

We conduct experiments on three multi-omics datasets across tissues and scales, covering challenges such as high-dimensional protein targets, batch effects, and data sparsity.

**Standard benchmark.** Following (Hu et al., 2024), we evaluate scChord on the GSE100866 benchmark using RMSE, PCC, and CMD. Reported values are averaged over five runs. As shown in Table 1, scChord effectively balances prediction accuracy and biological fidelity. Detailed results are provided in Appendix C.1.

*Table 1.* Performance comparison on the **GSE100866** dataset. The best results are highlighted in **bold**, and the second-best results are underlined.

| Method | PCC-P ↑ | PCC-C ↑ | CMD-P ↓ | CMD-C ↓ | RMSE ↓ |
|---|---|---|---|---|---|
| cTP-net | 0.6168 | 0.4594 | 0.3768 | 0.0840 | 0.6018 |
| Seurat | 0.4053 | 0.8315 | 0.4037 | 0.0909 | 0.9895 |
| Dengkw | 0.7270 | 0.8869 | 0.0298 | 0.1262 | 0.8182 |
| LIGER | 0.6297 | 0.9356 | 0.0829 | 0.0433 | 0.5709 |
| scArches | 0.7056 | 0.8848 | 0.0236 | 0.1157 | 0.8047 |
| sciPENN | 0.8305 | 0.8241 | 0.0037 | 0.1960 | 0.5538 |
| scVAEIT | 0.8525 | 0.9295 | 0.2543 | 0.0758 | 0.7716 |
| totalVI | 0.6629 | 0.8014 | 0.1077 | 0.1491 | 0.9779 |
| **scChord** | **0.8655** | **0.9392** | **0.0029** | **0.0316** | **0.5384** |

In terms of accuracy, scChord achieves the lowest RMSE and highest protein-level correlation (PCC-P). Unlike deterministic regression models (e.g., cTP-net) that minimize error by predicting conditional means, often leading to variance collapse, scChord recovers the full distribution of protein abundances. This capability is critical for resolving subtle biological heterogeneity typically smoothed out by mean-seeking approaches.

Crucially, scChord demonstrates superior structural fidelity, attaining the lowest Correlation Matrix Distance (CMD) scores at both protein and cellular levels. The order-of-magnitude improvement in CMD-P specifically indicates that our flow-based framework correctly learns the underlying co-expression patterns rather than treating proteins as independent variables. By aligning the manifold geometry of the predicted data with biological ground truth, scChord preserves the complex distributional dependency of the target proteome.

**Generalization to Large-Scale Data.** To further assess the generalizability of scChord, we evaluate its performance on the large-scale GSE164378-3P dataset (161,764 cells, 228 proteins). Reported values are averaged over five independent runs. As shown in Table 2, scChord achieves superior results across all metrics, demonstrating strong scalability and generalization to complex multimodal atlases. Detailed results are provided in Appendix C.1.

*Table 2.* Performance comparison on the **GSE164378-3P** dataset.

| Method | PCC-P ↑ | PCC-C ↑ | CMD-P ↓ | CMD-C ↓ | RMSE ↓ |
|---|---|---|---|---|---|
| cTP-net | 0.4033 | 0.2884 | 0.6543 | 0.1424 | 1.0236 |
| Dengkw | 0.5171 | 0.8744 | 0.1791 | 0.1024 | 1.0655 |
| LIGER | 0.4628 | 0.8678 | **0.1049** | 0.1009 | 1.0254 |
| scArches | 0.4400 | 0.8893 | 0.2725 | 0.0597 | 1.0405 |
| sciPENN | 0.5243 | 0.5646 | 0.1678 | 0.1363 | 0.7840 |
| scVAEIT | 0.5132 | 0.8887 | 0.4307 | 0.1466 | 1.0256 |
| totalVI | 0.5169 | **0.9243** | 0.1722 | 0.0529 | 1.0257 |
| **scChord** | **0.5274** | 0.8934 | 0.1690 | **0.0033** | **0.7779** |

Experimental results reveal that the scalability of deterministic regression methods like sciPENN and cTP-net significantly declines on large-scale datasets. As data volume increases, probabilistic modeling methods such as totalVI

and scVAEIT demonstrate growing competitiveness in this non-few-shot scenario, though their precision still requires improvement. LIGER employs the iNMF algorithm to align protein manifolds by prioritizing the extraction of shared signals, but this aggressive forced alignment strategy often sacrifices the preservation of fine-grained biological structures and data-specific details. In contrast, scChord achieves the best or second-best results across all metrics and establishes a substantial lead in reconstruction precision, demonstrating its robust scalability on complex multimodal atlases and its ability to preserve cell-level biological heterogeneity.

**Robustness Analysis.** To rigorously evaluate the method'stransferability across unseen biological contexts, we performed leave-one-group-out experiments on the GSE164378-5P ECCITE-seq dataset. Following scTranslator's protocol (Liu et al., 2025), we evaluated generalization across three axes: **Cell Type** (unseen states), **Batch** (cross-donor), and **Time** (temporal shifts).

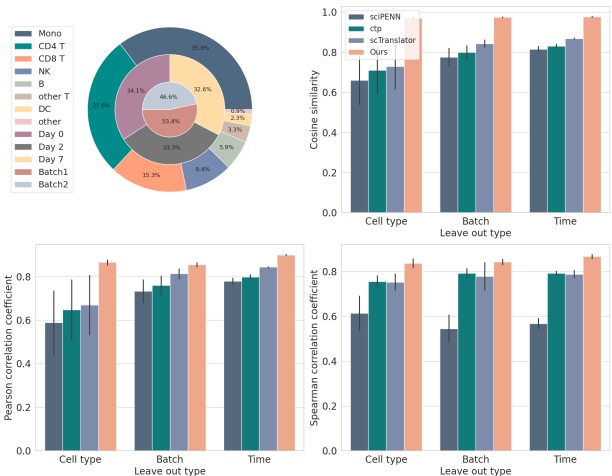

*Figure 4.* Generalization performance on the ECCITE-seq dataset.

As illustrated in Figure 4, the dataset exhibits marked imbalance in cell type distribution while maintaining relatively balanced batch and time point distributions (top-left panel). Despite these challenges, scChord consistently outperforms state-of-the-art baselines.

In the **Cell Type** hold-out scenario, scChord achieves superior cosine similarity and correlation coefficients, surpassing the foundation model scTranslator. This indicates that our manifold rectification strategy successfully captures intrinsic gene-protein distributional dependency that generalizes beyond specific cell identities seen during training.

Similarly, in **Batch** and **Time** hold-out settings, scChord demonstrates remarkable stability, maintaining high performance where regression-based methods often falter due to batch effects. This robustness confirms that scChord learns invariant biological mechanisms rather than overfitting to

batch-specific technical noise.

## 4.2. Mechanism Analysis: Decoding Geometric Rectification

To connect the theory in Section 3.1 and Appendix A.1 with practice, we analyze how rectification changes the geometry of the learned latent space. Standard metric comparisons are available in Section 4.3 (Variant D).

We compare scChord's rectified representation to an unrectified baseline trained with deterministic reconstruction, and evaluate (i) smoothness of decoded interpolations, (ii) solver difficulty during sampling, and (iii) encoder sensitivity to small input perturbations.

**Latent manifold interpolation.** We construct geodesic interpolation trajectories between latent cell embeddings $\mathbf{z}_1, \mathbf{z}_2$ by projecting linear interpolants onto the manifold via iterative encoder-decoder cycles, ensuring the path remains on the learned manifold. Decoding these trajectories to protein space reveals the phenotypic landscape.

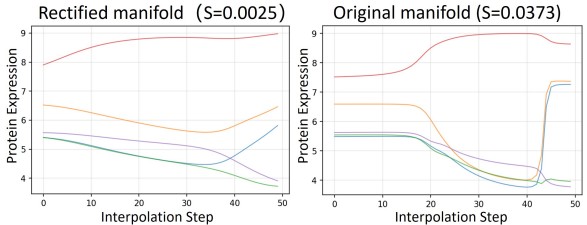

*Figure 5.* **Latent Space Interpolation Trajectories.** Interpolating between cell states reveals the learned manifold structure.

As visualized in Figure 5, the original manifold yields highly oscillatory and non-monotonic protein trajectories. This phenomenon quantitatively manifests as a high second-order difference score

$$\mathcal{S} = \frac{1}{M(T-2)} \sum_t |\hat{y}_{t+1} - 2\hat{y}_t + \hat{y}_{t-1}|, \qquad (8)$$

confirming that the unrectified latent space is riddled with local irregularities caused by overfitting to uncertainty. Conversely, the rectified manifold demonstrates smooth biological transitions, supporting our hypothesis that incorporating noise parameters $(\theta, \pi)$ allows the latent variable to capture the underlying continuous biological manifold.

**Geometric stiffness and encoder stability.** We quantify how the learned geometry affects downstream transport with two complementary measures: the number of function evaluations (NFE) required by an adaptive ODE solver during sampling, and the empirical local sensitivity of the encoder.

*ODE solver cost (NFE).* We track the computational cost $N_{\text{NFE}}$ incurred by the adaptive solver (Dopri5) during sam-

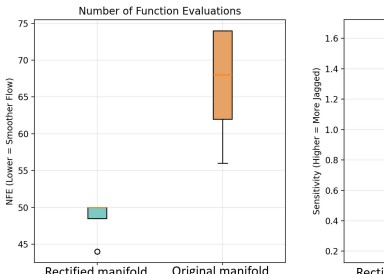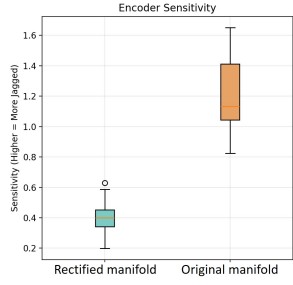

*Figure 6.* **Geometric Stiffness and Stability Metrics.** Box plots illustrating the distribution of Number of Function Evaluations (NFE) and Encoder Sensitivity ($\bar{L}$) across the test set. The Rectified Manifold demonstrates significantly lower NFE and Sensitivity compared to the unrectified baseline, with reduced variance indicating superior training stability. This confirms that rectification mitigates ODE stiffness and effectively constrains the local Lipschitz constant.

pling. When target trajectories are locally irregular, the learned vector field becomes stiff, forcing the solver to take smaller steps to satisfy error tolerances. Rectification substantially reduces the median NFE (Figure 6, left), indicating smoother dynamics and easier numerical integration.

*Encoder sensitivity.* We interpret encoder sensitivity as the empirical local Lipschitz constant of the encoding map with respect to the observation $y$. This metric directly reflects the *roughness* of the induced target distribution: larger constants indicate that the target representation varies abruptly under small measurement perturbations, implying a highly irregular effective geometry (see Appendix A.1). Concretely, we estimate the sensitivity $Lip_{\text{enc}}(y)$ as:

$$Lip_{\text{enc}}(y) \;\approx\; \frac{\|\Delta\mu_{\text{enc}}(y)\|}{\|\Delta y\|}, \tag{9}$$

where $\Delta y$ is a small random perturbation and $\Delta\mu_{\text{enc}}(y)$ is the corresponding shift in the posterior mean. Empirically, the unrectified baseline exhibits substantially larger $Lip_{\text{enc}}$ in noisy regions, whereas scChord reduces this constant by absorbing measurement noise into the likelihood, thereby smoothing the effective target manifold.

Beyond interpolation smoothness and ODE solver cost, we further quantified latent geometry using spectral, diffusion, eigenspace, and intrinsic-dimensionality metrics (detailed in Appendix D). The rectified manifold consistently improves nonlinear geometric regularity, supporting that Stage 1 acts as a geometric preconditioner rather than merely changing the reconstruction loss.

### 4.3. Ablation Studies

To dissect the contribution of each component in scChord, we conducted a comprehensive ablation study on the GSE100866 benchmark. The results, summarized in Table 3, highlight the critical role of our architectural choices.

**Ablation analysis.** We analyze Table 3 from a causal perspective: whether (i) rectifying protein geometry is a prerequisite for learning stable dynamics, (ii) probabilistic rectification and KL regularization play distinct roles in shaping the latent manifold, and (iii) consistency regularization improves robustness under severe RNA sparsity.

*Two-stage coupling.* Removing either stage degrades performance, with distinct failure modes. Without Stage 1 (Variant A), the model learns dynamics on raw protein counts; over-dispersion and dropout yield a rugged, high-curvature target set, reflected by a drop in PCC-P ($0.3572$) and an increase in RMSE ($1.1043$). Without Stage 2 (Variant B), the model loses RNA-conditioned flow matching; while PCC-C stays high, CMD and error worsen, so rectification alone cannot substitute for conditional transport.

*Probabilistic rectification vs. deterministic formulation.* Compared with deterministic variants (C–E), probabilistic rectification (Full model, F, G) achieves better accuracy and structural fidelity by modeling count noise in the likelihood. Deterministic training (Variant C) can achieve moderate correlation, but it remains constrained to point-estimation behavior and is more susceptible to encoding measurement noise into the latent geometry.

A particularly revealing case is the *KL-only* regime, i.e., enforcing KL regularization in a deterministic setting (Variant E). Its catastrophic collapse reflects a structural incompatibility between (i) deterministic reconstruction preserving sample-specific high-frequency fluctuations and (ii) KL regularization contracting the posterior towards a Gaussian prior. Without a probabilistic decoder to absorb over-dispersion and zero-inflation, KL forces the model to "explain away" genuine variability by shrinking the latent code, yielding an overly concentrated posterior and losing discriminative geometry. Thus, KL is beneficial only when paired with an explicit noise model: probabilistic rectification provides a place (decoder distributional parameters) to store measurement uncertainty, so KL acts as geometric regularization rather than destructive contraction.

Consistently, removing KL under probabilistic rectification (Variant F) remains strong but is weaker than the full model, especially on structure-related metrics (CMD). This suggests that, once noise is absorbed by the likelihood, KL mainly contributes by shaping a compact, smooth latent geometry that supports stable flow learning and preserves correlation structure.

*Robustness via transcriptomic consistency.* Finally, removing the consistency term (Variant G) increases RMSE and worsens CMD, showing it is essential. By requiring the conditional vector fields induced by full and corrupted transcriptomic conditions to agree, the model becomes less sensitive to dropout-driven perturbations in RNA, reducing

*Table 3.* **Component-wise ablation study on GSE100866 (three-level view).** We organize variants by (i) *end-to-end architecture* (two-stage coupling), (ii) *mechanistic formulation* (probabilistic vs. deterministic latent learning and KL regularization), and (iii) *robustness* (transcriptomic consistency).

| ABLATION VARIANT | CONFIGURATION | | | KEY METRICS | | | | |
|---|---|---|---|---|---|---|---|---|
| | PARADIGM | KL REG. | $\mathcal{L}_{\text{CONS}}$ | PCC-P ↑ | PCC-C ↑ | CMD-P ↓ | CMD-C ↓ | RMSE ↓ |
| *(I) two-stage coupling* | | | | | | | | |
| (A) W/O STAGE1 | RAW SPACE | - | - | 0.3572 | 0.8006 | 0.0689 | 0.0150 | 1.1043 |
| (B) W/O STAGE2 | RAW SPACE | - | - | 0.2066 | 0.9080 | 0.3285 | 0.0480 | 1.1725 |
| *(II) probabilistic vs. deterministic* | | | | | | | | |
| (C) BASELINE | DETERMINISTIC | × | × | 0.7167 | 0.8712 | 0.0349 | 0.0587 | 0.8563 |
| (D) W/O PROB&KL | DETERMINISTIC | × | ✓ | 0.7788 | 0.9021 | 0.0152 | 0.0485 | 0.7106 |
| (E) W/O PROB | DETERMINISTIC | ✓ | ✓ | 0.0230 | 0.4853 | 0.2599 | 0.2361 | 2.2322 |
| (F) W/O KL | PROBABILISTIC | × | ✓ | 0.8414 | 0.9295 | 0.0031 | 0.0341 | 0.5853 |
| *(III) transcriptomic consistency* | | | | | | | | |
| (G) W/O $\mathcal{L}_{\text{CONS}}$ | PROBABILISTIC | ✓ | × | 0.8506 | 0.9251 | 0.0076 | 0.0456 | 0.6220 |
| **SCCHORD (FULL)** | **PROBABILISTIC** | ✓ | ✓ | **0.8655** | **0.9392** | **0.0029** | **0.0316** | **0.5384** |

prediction drift under realistic sparse inputs.

## 4.4. Visualization of Biological Fidelity

While global metrics like RMSE quantify average error, they often fail to capture the intrinsic stochasticity and phenotypic heterogeneity of single-cell data. To rigorously evaluate distributional alignment, we employ the Kolmogorov-Smirnov (KS) distance to measure the discrepancy between predicted and ground-truth distributions. We also visualize two markers with distinct properties: bimodal CD4 and heavy-tailed CD45RA. Detailed KS statistics and extended visualizations for all markers are in Appendix C.2.

Beyond distributional alignment, we also report appendix-level statistics on mean, variance, and noise preservation. These results show that while many methods can match mean expression, they differ substantially in variance and noise fidelity, and scChord better preserves stochastic variability rather than collapsing it.

**Recovery of Multimodal Populations (CD4).** The surface protein CD4 exhibits a classic bimodal distribution, reflecting the biological segregation between T-helper cells (high abundance) and other cell types. As shown in Figure 7(a), regression-based approaches such as sciPENN and cTP-net exhibit a distinct tendency to predict the conditional mean, failing to match the true dynamic range. Similarly, VAE-based modeling methods like totalVI struggle to precisely recover the specific density profile of CD4. In contrast, scChord remarkably recapitulates this bimodality, with its predicted density (Red) aligning closely with the ground-truth kernel density estimate (Black dashed line).

**Preservation of Continuous Spectrums (CD45RA).** We further evaluate performance on CD45RA, a marker characterized by a broad, continuous dynamic range associated

with memory T-cell differentiation. Figure 7(b) demonstrates that scChord faithfully recovers the entire spectrum of expression, including the heavy-tailed regions that indicate high-expression rare cells. Competing methods reveal severe limitations here: Dengkw and scArches suffer from mode collapse (visible as spurious spikes at zero or low values), while Seurat produces a contracted distribution that misses the biological variance. scChord's superior alignment across the full support of the distribution validates the efficacy of our *Probabilistic Manifold Rectification*, which successfully decouples technical noise from the biological signal to prevent the collapse of intrinsic heterogeneity.

## 4.5. Computational Complexity and Runtime

*Table 4.* Computational complexity and runtime at 100k cells.

| Method | Train (s) | CPU Mem (MB) | GPU Mem (MB) | Infer (s/200) |
|---|---|---|---|---|
| cTP-net | — | — | — | 0.259 |
| sciPENN | — | — | — | 0.002 |
| Dengkw | 195 | 4,910 | 0 | 0.023 |
| Liger | 403 | 36,356 | 0 | 0.026 |
| scArches | 1,419 | 23,779 | 833 | 0.069 |
| scVAEIT | 2,923 | 9,968 | 712 | 0.465 |
| totalVI | 1,297 | 22,140 | 392 | 0.030 |
| **scChord** | 1,636 | 10,766 | 229 | 0.488 |

*Note.* cTP-net and sciPENN training/memory metrics are unavailable because both methods exceeded 64 GB RAM and crashed at the 100k-cell scale.

We benchmarked all models at 100k cells on GSE164378-3P (NVIDIA RTX 4080), Table 4 summarizes results at 100k cells. Non-deep-learning methods (Dengkw, Liger) train much faster, which is an expected trade-off for lower modeling capacity. Among deep generative models, scChord (1,636s; Stage 1: 576s, Stage 2: 1,061s) is competitive (1.8× faster than scVAEIT; comparable to totalVI). We also acknowledge a key limitation: scChord has the highest inference latency due to multi-step ODE integration

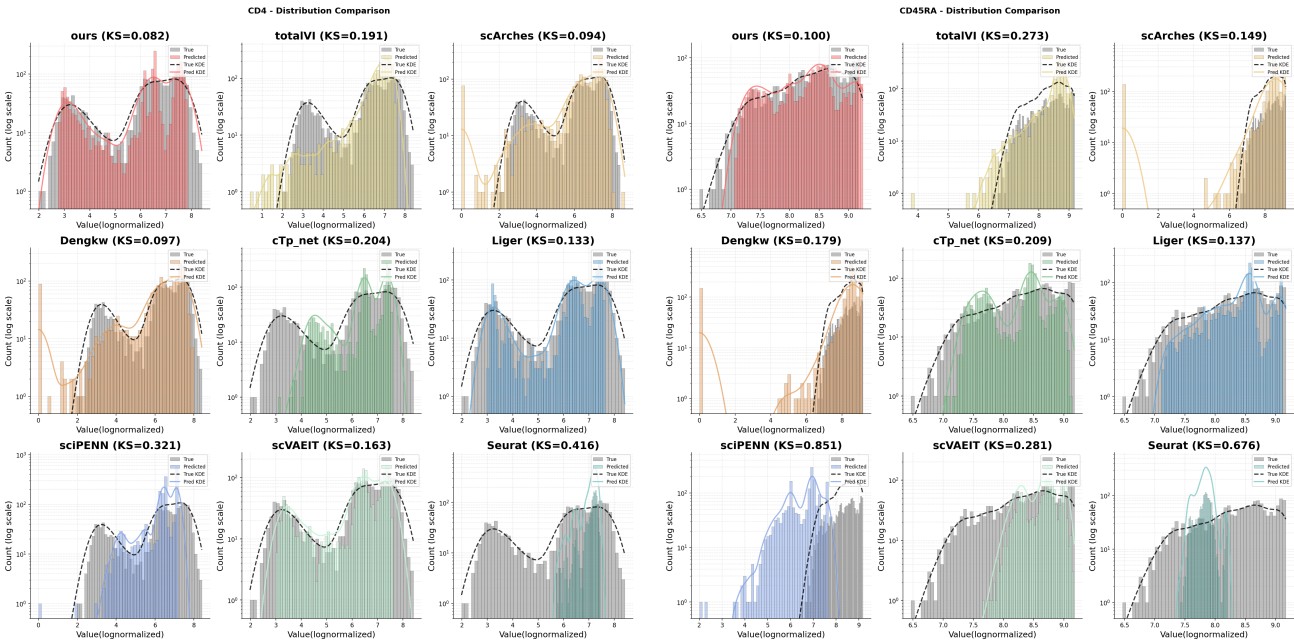

*Figure 7.* **Distributional fidelity of predicted protein expression.** (a) CD4 exhibits a bimodal distribution. (b) CD45RA has a broad, heavy-tailed spectrum. Grey bars/black dashed KDE represent ground truth; red represents scChord.

with `dopri5`. Preliminary tests show that fewer integration steps and fixed-step solvers (e.g., Euler/midpoint) can substantially reduce latency with only marginal accuracy loss. In contrast, scChord has the lowest GPU peak memory (229 MB) and moderate CPU memory (∼10.7 GB at 100k), improving accessibility.

## 5. Conclusion

In this work, we introduced scChord to address the challenge of rough latent geometry in cross-modal translation, where biological sparsity and over-dispersion typically destabilize continuous dynamics. By synergizing Probabilistic Manifold Rectification with Consistency-Regularized Flow Matching, our framework effectively disentangles uncertainty from the latent biological structure. Theoretical and empirical analyses demonstrate that this geometric regularization mitigates ODE stiffness and ensures high-fidelity reconstruction. Consequently, scChord achieves state-of-the-art performance on CITE-seq benchmarks, demonstrating a superior capability to reconstruct complex, multimodal stochastic landscapes essential for resolving fine-grained cellular heterogeneity.

Despite these advances, preserving protein-level structural fidelity remains a challenge as the dimensionality of the target proteome expands. We observed that the Correlation Matrix Distance for proteins (CMD-P) tends to increase as the number of target proteins rises, suggesting that implic-

itly learning intricate inter-protein co-expression patterns becomes increasingly difficult in high-dimensional spaces without structural guidance. Addressing this limitation may require incorporating pathway-informed graph priors or hierarchical latent structures. These additions would explicitly model complex dependencies, ensuring that the global topology of the proteome is preserved even in large-scale output spaces.

## Impact Statement

The presented work deals with the fundamental geometric irregularities inherent in stochastic multi-omics data and studies how probabilistic manifold rectification enables high-fidelity cross-modal translation. We envision the release of scChord as an accessible, open-source framework to democratize robust protein inference from standard RNA sequencing. Dealing with biological data, scChord could be applied in sensitive clinical settings for disease profiling, requiring strict adherence to privacy standards regarding patient information.

## Code accessibility

The training code and related visualization programs for scChord are available at https://github.com/Jave-Zhang/scChord to facilitate reproducibility while maintaining the double-blind review protocol.

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

# A. Theory Appendix

## A.1. Geometric Irregularity as a Sensitivity Lower Bound

This appendix formalizes the key geometric obstruction discussed in Section 3.1. We show that under non-degenerate observation noise, any deterministic representation map defined on a continuous relaxation of the data space that separates distinct measurement realizations must exhibit high local sensitivity, yielding a "rough" or irregular geometry.

**Definition A.1** (Local sensitivity). Let the encoder $g : \mathbb{R}^{d_y} \to \mathbb{R}^m$ be a differentiable function defined on the continuous embedding of the observation space. We define the local sensitivity at $y$ as the spectral norm of its Jacobian:

$$\mathsf{Sens}(g; y) \triangleq \|\nabla_y g(y)\|_{\mathrm{op}} = \sup_{u:\|u\|=1} \|\nabla_y g(y) \cdot u\|. \tag{10}$$

**Assumption A.2** (Continuous relaxation and non-degenerate noise). We assume the discrete observations are embedded in a continuous space $\mathcal{Y} = \mathbb{R}^{d_y}$. For each condition $x \in \mathcal{X}$, observations satisfy $y = f(x) + \xi$ with $\mathbb{E}[\xi \mid x] = 0$ and $\mathrm{tr}(\Sigma(x)) > 0$, where $\Sigma(x) = \mathrm{Var}(\xi \mid x)$ denotes the noise covariance.

**Assumption A.3** (Non-collapse of distinct realizations). There exist a separation margin $\delta > 0$ and a probability $p_0 \in (0, 1)$ such that for a given condition $x$, if $y_1, y_2 \stackrel{\text{i.i.d.}}{\sim} p(\cdot \mid x)$ are two independent realizations, then the encoder preserves their distinction with high probability:

$$\mathbb{P}\big(\|g(y_1) - g(y_2)\| \geq \delta\big) \geq p_0. \tag{11}$$

This assumption captures the behavior of a deterministic model fitting the noise.

**Proposition A.4** (Sensitivity lower bound implies geometric roughness). *Under the above assumptions, with probability at least $p_0$ over the draw of $(y_1, y_2)$, there exists a point $\bar{y}$ lying on the line segment $[y_1, y_2]$ connecting the observations such that the local sensitivity is lower-bounded by the ratio of output separation to input noise:*

$$\mathsf{Sens}(g; \bar{y}) \geq \frac{\delta}{\|y_1 - y_2\|}. \tag{12}$$

*Consequently, as inputs $y_1, y_2$ become closer (governed by $\Sigma(x)$) while their embeddings remain separated by $\delta$, the local Lipschitz constant $\mathsf{Sens}(g; \bar{y})$ grows arbitrarily large.*

*Proof sketch.* Consider the event where $\|g(y_1) - g(y_2)\| \geq \delta$. Let $\gamma(t) = (1 - t)y_1 + ty_2$ for $t \in [0, 1]$ parameterize the line segment connecting $y_1$ and $y_2$. By the fundamental theorem of calculus for vector-valued functions, we express the difference as:

$$g(y_2) - g(y_1) = \int_0^1 \nabla_y g\big(\gamma(t)\big) \cdot (y_2 - y_1) \, \mathrm{d}t. \tag{13}$$

Applying the triangle inequality for integrals and the definition of the operator norm $\|\cdot\|_{\mathrm{op}}$:

$$\|g(y_2) - g(y_1)\| \leq \int_0^1 \big\|\nabla_y g\big(\gamma(t)\big)\big\|_{\mathrm{op}} \|y_2 - y_1\| \, \mathrm{d}t. \tag{14}$$

Let $\phi(t) = \|\nabla_y g(\gamma(t))\|_{\mathrm{op}}$. The inequality simplifies to:

$$\delta \leq \|g(y_2) - g(y_1)\| \leq \|y_2 - y_1\| \int_0^1 \phi(t) \, \mathrm{d}t. \tag{15}$$

By the Mean Value Theorem for integrals, there exists $t^* \in [0, 1]$ such that $\phi(t^*) = \int_0^1 \phi(t) dt$. However, strictly speaking, for the inequality to hold, we only need the existence of a point where the integrand is at least the average value. Thus, there exists $\bar{y} = \gamma(t^*)$ such that:

$$\mathsf{Sens}(g; \bar{y}) = \phi(t^*) \geq \int_0^1 \phi(t) dt \geq \frac{\|g(y_2) - g(y_1)\|}{\|y_2 - y_1\|} \geq \frac{\delta}{\|y_2 - y_1\|}. \tag{16}$$

$\square$

**Corollary A.5** (Implication for flow stiffness)**.** *Suppose Stage 2 trains a flow matching model with regression targets* $x_1 = g(y)$. *The proposition implies that a deterministic encoder g fitting noisy data will possess large local Lipschitz constants (*$\mathsf{Sens}(g; \cdot)$*) in high-noise regions. Consequently, small variations in the input observation y result in large displacements in the target* $x_1$. *This increases the variance of the flow velocity field* $\nabla_z v_\theta$, *empirically correlating with higher curvature trajectories and increased NFE (Number of Function Evaluations) for ODE solvers. Stage 1 probabilistic rectification mitigates this by absorbing the noise* $\xi$ *into the decoder likelihood, allowing g(y) (as the posterior mean) to represent the smooth biological state rather than the noisy realization.*

## A.2. Theoretical Analysis of Geometric Rectification

In this section, we provide a formal derivation of Geometric Rectification, analyzing how the loss function dictates the geometry of the latent manifold through gradient dynamics. We define the **Gradient Stiffness** $S(\mathcal{L})$ as the magnitude of the gradient with respect to the pre-activation output (mean $\mu$) per unit of prediction error. A high stiffness implies that the model is sensitive to residuals, forcing the latent variable $z$ to track noise.

**Baseline: Isotropic Stiffness of Gaussian Loss.**    For a standard Gaussian likelihood (MSE objective) $\mathcal{L}_{\mathrm{MSE}} = \frac{1}{2}(y - \mu)^2$, the gradient $\nabla_\mu \mathcal{L}_{\mathrm{MSE}} = -(y - \mu)$ yields a constant stiffness $S_{\mathrm{MSE}} = 1$. The gradient magnitude is linearly unbounded with respect to the residual $|y - \mu|$. In the presence of outliers or heavy-tailed noise, this generates large gradient norms $\|\nabla_z \mathcal{L}\| = \|y - \mu\| \cdot \|\nabla_z \mu\|$. To minimize this loss, the encoder must adjust $z$ significantly to reduce the residual, thereby encoding noise into the latent coordinates and increasing the local Lipschitz constant $\|\nabla_y z\|$.

**Mechanism 1: Inverse-Variance Scaling in Negative Binomial.**    The Negative Binomial (NB) loss features a variance $\sigma^2(\mu) = \mu + \alpha\mu^2$, where $\alpha = 1/\theta$ is the dispersion. The gradient of the NLL w.r.t. $\mu$ follows the score function of a Generalized Linear Model (GLM):

$$\nabla_\mu \mathcal{L}_{\mathrm{NB}} = -\frac{y - \mu}{\sigma^2(\mu)} = -\underbrace{\left(\frac{1}{1 + \alpha\mu}\right)}_{\lambda(\mu, \alpha)} \cdot \frac{y - \mu}{\mu}. \tag{17}$$

The term $\lambda(\mu, \alpha)$ acts as a **Variance-Adaptive Scaling Factor**:

- **High-Expression Suppression:** As $\mu \to \infty$, $\lambda \to 0$. The gradient contribution is naturally suppressed by the quadratic variance term.

- **Heteroscedasticity Handling:** For highly over-dispersed genes (large $\alpha$), $\lambda$ decreases, ensuring that observations with high inherent uncertainty contribute less to the displacement of the latent variable $z$.

**Mechanism 2: Posterior-Weighted Suppression in ZINB.**    For Zero-Inflated data, we consider observing $y = 0$. The likelihood is the mixture $p(y = 0) = \pi + (1 - \pi)p_{\mathrm{NB}}(0|\mu, \alpha)$. The gradient of $\mathcal{L}_{\mathrm{ZINB}} = -\log p(y = 0)$ w.r.t. $\mu$ is:

$$\nabla_\mu \mathcal{L}_{\mathrm{ZINB}}\Big|_{y=0} = \underbrace{\left[\frac{(1 - \pi)p_{\mathrm{NB}}(0)}{\pi + (1 - \pi)p_{\mathrm{NB}}(0)}\right]}_{\gamma(z, y=0)} \cdot \nabla_\mu \mathcal{L}_{\mathrm{NB}}\Big|_{y=0}. \tag{18}$$

Here, $\gamma(z, y = 0) = P(\text{State} = \text{Biological} \mid y = 0)$ represents the **Posterior Signal Confidence**. This introduces a dynamic suppression mechanism: as $\pi \to 1$, $\gamma \to 0$ and the gradient vanishes. Unlike MSE, where a zero observation ($y = 0$) for a high-expression cell ($\mu \gg 0$) generates a massive penalty, ZINB allows the gradient to vanish if the model assigns the event to technical dropout.

**Conclusion: Lipschitz Boundedness.**    The effective gradient propagated to the encoder is $\nabla_z \mathcal{L} = \gamma(z) \cdot \lambda(\mu, \alpha) \cdot \frac{y - \mu}{\mu} \cdot \nabla_z \mu$. In regions dominated by noise (dropout or high dispersion), the scaling coefficients $\gamma$ and $\lambda$ tend to 0. This makes the encoder update $\Delta\phi \propto \nabla_z \mathcal{L}$ less sensitive to noisy observations $y$ and empirically encourages smoother latent geometry in $\mathcal{M}_{\mathrm{lat}}$.

## B. Experimental Details

### B.1. Datasets

We utilize three diverse CITE-seq and ECCITE-seq datasets to evaluate model performance across different biological contexts. The detailed statistics are summarized in Table 5. For all baseline methods, the datasets were split into training and test sets using an 0.8:0.2 ratio with a fixed random seed of 0. Furthermore, all RMSE metrics were calculated in log-normalized space to ensure comparability across different scales and methods.

1. **GSE100866 (Human CBMC):** A baseline CITE-seq dataset profiling Cord Blood Mononuclear Cells (Stoeckius et al., 2017). It contains 8,617 cells with paired measurements of 36,280 RNA genes and 13 surface proteins. This dataset exhibits high transcriptomic sparsity (96%) and serves as a standard benchmark for method development.

2. **GSE164378 (Human PBMC):** A large-scale massive multimodal atlas (Hao et al., 2024) comprising two distinct batches: the 3-prime (3P) subset and the 5-prime (5P) ECCITE-seq subset. Together, they consist of 161,764 cells with a high-dimensional antibody panel of 228 proteins. The inclusion of both batches allows us to assess the model's scalability on complex atlases (3P) and its generalization capability through leave-one-batch-out validation (5P).

*Table 5.* Summary of CITE-seq datasets used in this study. The collection covers human and mouse tissues, ranging from small-scale panels to high-throughput multimodal atlases.

| DATASET | SPECIES | TISSUE | BATCHES | CELLS | GENES | PROTEINS | RNA SPARSITY | PROTEIN SPARSITY |
|---|---|---|---|---|---|---|---|---|
| GSE100866 | HUMAN | CBMC | 1 | 8,617 | 36,280 | 13 | 96% | 0% |
| GSE164378 (3P) | HUMAN | PBMC | 2 | 161,764 | 33,538 | 228 | 93% | 9% |
| GSE164378 (5P) | HUMAN | PBMC | 2 | 49,147 | 33,538 | 54 | 96% | 12% |

### B.2. Baselines

To comprehensively evaluate scChord, we benchmark against a diverse set of state-of-the-art methods: (1) *Deterministic Translation*: **cTP-net**(Zhou et al., 2020), **sciPENN**(Lakkis et al., 2022), and **Guanlab-dengkw**(Lance et al., 2022); (2) *Probabilistic Generative Models*: **totalVI**(Gayoso et al., 2021) and **scVAEIT**(Du et al., 2022); and (3) *Integration Frameworks*: **Seurat v5**(Hao et al., 2024), **LIGER**(Welch et al., 2019), and **scArches**(Lotfollahi et al., 2022). Additionally, we include the foundation model **scTranslator**(Liu et al., 2025) to assess zero-shot generalization capabilities. Since **scTranslator** is a foundation model pretrained on most existing datasets, we validate its performance on the specific test datasets designated in the original publication to prevent data leakage, adopting the metrics directly from their official open-source results. We do not include **scDM**(Yu et al., 2024) in our comparison as its source code is not publicly available and its reported performance is not competitive compared to the baselines successfully reproduced in this study. Detailed implementation settings for these baselines are as follows:

1. **totalVI**: We followed the official tutorial settings (`https://docs.scvi-tools.org/en/stable/tutorials/notebooks/multimodal/cite_scrna_integration_w_totalVI.html`). We set `latent_distribution = 'normal'` and `n_layers_decoder = 2`.

2. **scArches**: We followed the totalVI surgery pipeline on the scArches documentation. The model was trained with `epochs = 200`, `weight_decay = 0.0`, and default `plan_kwargs`.

3. **Guanlab-dengkw**: We followed the tutorial provided in the NeurIPS 2021 multimodal challenge top methods repository.

4. **sciPENN**: Following the sciPENN GitHub repository, we used the parameters: `n_epochs = 10,000`, `ES_max = 12`, `decay_max = 6`, `decay_step = 0.1`, and `lr = 10^{-3}`.

5. **cTP-net**: We followed the instructions on the cTP-net repository, setting `n_batches = 32` and `max_epochs = 4`.

6. **Seurat**: We utilized Seurat v5 (`https://satijalab.org/seurat/`) with `reduction = 'cca'` and the `TransferData` function. For multimodal integration, Weighted Nearest Neighbor (WNN) analysis was performed.

7. **LIGER**: We followed the LIGER tutorial and applied the `imputeKNN` function with `norm = FALSE` and `scale = FALSE`.

8. **scVAEIT**: We followed the scVAEIT GitHub repository guidelines; predictions were obtained using the `get_recon` function.

### B.3. Implementation Details

All models were implemented in PyTorch and trained on NVIDIA RTX 4080 GPUs using the AdamW optimizer. For preprocessing, we selected the top 1,000 highly variable genes.

In Stage 1 (ProteinVAE), we trained for 600 epochs with a batch size of 512, a learning rate of $2 \times 10^{-4}$, a latent dimension $d_z = 32$, and $\beta_{\mathrm{KL}} = 0.8$. In Stage 2 (CFM), the model was trained for 200 epochs with a batch size of 512, a learning rate of $1 \times 10^{-4}$, and a condition embedding dimension $d_c = 512$. Regularization parameters were set to $p_{\mathrm{uncond}} = 0.2$ and $\lambda_{\mathrm{cons}} = 0.2$. For inference, we used the `dopri5` ODE solver with relative and absolute tolerances of $10^{-5}$, a CFG scale of $w = 3.0$, and 50 integration steps.

### B.4. Evaluation Metrics

1. **Pearson Correlation Coefficient (PCC)**: The PCC is defined by the following equation:

$$\mathrm{PCC}(\mathbf{x}, \mathbf{y}) = \frac{\sum_{i=1}^{n}(x_i - \bar{x})(y_i - \bar{y})}{\sqrt{\sum_{i=1}^{n}(x_i - \bar{x})^2}\sqrt{\sum_{i=1}^{n}(y_i - \bar{y})^2}} \tag{19}$$

   In this study, protein abundances are evaluated in the same log-normalized, per-protein z-scored space for all methods. PCC-P is computed protein-wise across the $N \times P$ protein matrix, i.e., each protein contributes one correlation across the $N$ cells and the resulting $P$ values are averaged. PCC-C is computed cell-wise across the same $N \times P$ matrix, i.e., each cell contributes one correlation across the $P$ proteins and the resulting $N$ values are averaged.

2. **Correlation Matrix Distance (CMD)**: CMD is used to measure the difference between two correlation matrices $\mathbf{R}_1$ and $\mathbf{R}_2$. A lower CMD value indicates a better result. The CMD is defined as:

$$d(\mathbf{R}_1, \mathbf{R}_2) = 1 - \frac{\mathrm{trace}(\mathbf{R}_1 \mathbf{R}_2)}{\|\mathbf{R}_1\|_F \|\mathbf{R}_2\|_F} \tag{20}$$

   where $\mathrm{trace}(\mathbf{R}_1 \mathbf{R}_2)$ represents the trace of matrix $\mathbf{R}_1 \times \mathbf{R}_2$ and $\| \cdot \|_F$ is the Frobenius norm of a matrix. CMD-P compares the $P \times P$ protein-protein correlation matrices computed across cells, while CMD-C compares the $N \times N$ cell-cell correlation matrices computed across proteins.

3. **Root Mean Square Error (RMSE)**: RMSE is used to quantify the difference between the predicted values ($\mathbf{X}$) and true values ($\hat{\mathbf{X}}$). Both the predicted and true values were normalized and rescaled using $z$-scores. The RMSE is mathematically defined as:

$$\mathrm{RMSE} = \sqrt{\frac{1}{NP} \sum_{i=1}^{N} \sum_{j=1}^{P} (\hat{y}_{ij} - y_{ij})^2} \tag{21}$$

   where $N$ is the number of cells, $P$ is the number of proteins, and the comparison is performed in the same log-normalized, per-protein z-scored evaluation space used for all baselines.

### B.5. Kernel Density Estimation (KDE)

Kernel Density Estimation is used to estimate the probability density function. We use a Gaussian kernel for estimation.

#### B.5.1. GAUSSIAN KERNEL DENSITY ESTIMATION FORMULA

For sample data $\mathbf{x} = \{x_1, x_2, \ldots, x_n\}$, the kernel density estimate at point $x$ is:

$$\hat{f}_h(x) = \frac{1}{n} \sum_{i=1}^{n} K_h(x - x_i) = \frac{1}{nh} \sum_{i=1}^{n} K\left(\frac{x - x_i}{h}\right) \tag{22}$$

where $K(\cdot)$ is the kernel function and $h$ is the bandwidth parameter.

### B.5.2. GAUSSIAN KERNEL FUNCTION

We use the Gaussian kernel (standard normal distribution):

$$K(u) = \frac{1}{\sqrt{2\pi}} \exp\left(-\frac{u^2}{2}\right) \tag{23}$$

Thus, the specific form of the Gaussian KDE is:

$$\hat{f}_h(x) = \frac{1}{nh\sqrt{2\pi}} \sum_{i=1}^{n} \exp\left(-\frac{(x - x_i)^2}{2h^2}\right) \tag{24}$$

### B.5.3. CONVERSION FROM DENSITY TO COUNTS

In visualization, we convert density values to counts to compare with histograms on the same scale:

$$\text{kde\_counts}(x) = \hat{f}_h(x) \times n \times \Delta x \tag{25}$$

where:

- $\hat{f}_h(x)$ is the KDE value at point $x$.

- $n$ is the sample size (len(true_protein) or len(pred_protein)).

- $\Delta x = \frac{x_{\max} - x_{\min}}{b}$ is the bin width of the histogram, where $b$ is the number of bins ($b = 50$ in our code).

### B.6. Kolmogorov-Smirnov Distance (KS Distance)

The KS distance measures the difference between two sample distributions, defined as the maximum difference between two empirical cumulative distribution functions (ECDF).

### B.6.1. EMPIRICAL CUMULATIVE DISTRIBUTION FUNCTION

For a sample $\mathbf{x} = \{x_1, x_2, \ldots, x_n\}$, the ECDF is defined as:

$$F_n(x) = \frac{1}{n} \sum_{i=1}^{n} \mathbf{1}_{x_i \leq x} = \frac{\text{number of observations} \leq x}{n} \tag{26}$$

where $\mathbf{1}_{x_i \leq x}$ is the indicator function:

$$\mathbf{1}_{x_i \leq x} = \begin{cases} 1 & \text{if } x_i \leq x \\ 0 & \text{otherwise} \end{cases} \tag{27}$$

### B.6.2. KS STATISTIC

For two samples $\mathbf{x}_{\text{true}} = \{x_1^{\text{true}}, \ldots, x_m^{\text{true}}\}$ and $\mathbf{x}_{\text{pred}} = \{x_1^{\text{pred}}, \ldots, x_n^{\text{pred}}\}$, the KS statistic is defined as:

$$D_{m,n} = \sup_{x \in \mathbb{R}} |F_m^{\text{true}}(x) - F_n^{\text{pred}}(x)| \tag{28}$$

where:

- $F_m^{\text{true}}(x)$ is the ECDF of the true data.

- $F_n^{\text{pred}}(x)$ is the ECDF of the predicted data.

- sup denotes the supremum (maximum value).

## C. Supplementary Experimental Results

### C.1. Statistical analysis of method comparisons

To evaluate the robustness and reproducibility of our results, all baseline methods were tested in five independent experiments using random seeds 0, 10, 20, 30, and 40. The quantitative metrics reported in this section reflect the aggregate performance across these runs. For qualitative visualizations throughout the appendix, we present results from one representative experimental instance.

*Table 6.* Replication results on the GSE100866 dataset. Performance metrics are reported as mean $\pm$ standard deviation across $n = 5$ independent initializations.

| METHOD | $n$ | PCC-P | PCC-C | CMD-P | CMD-C | RMSE |
|---|---|---|---|---|---|---|
| CTP-NET | 5 | $0.6168 \pm 0.0119$ | $0.4594 \pm 0.0126$ | $0.3768 \pm 0.0143$ | $0.0840 \pm 0.0060$ | $0.6018 \pm 0.0250$ |
| SEURAT | 5 | $0.4053 \pm 0.0058$ | $0.8315 \pm 0.0061$ | $0.4037 \pm 0.0110$ | $0.0909 \pm 0.0084$ | $0.9895 \pm 0.0150$ |
| DENGKW | 5 | $0.7270 \pm 0.0314$ | $0.8869 \pm 0.0057$ | $0.0298 \pm 0.0059$ | $0.1262 \pm 0.0111$ | $0.8182 \pm 0.0064$ |
| LIGER | 5 | $0.6297 \pm 0.0155$ | $0.9356 \pm 0.0038$ | $0.0829 \pm 0.0133$ | $0.0433 \pm 0.0037$ | $0.5709 \pm 0.0106$ |
| SCARCHES | 5 | $0.7056 \pm 0.0074$ | $0.8848 \pm 0.0108$ | $0.0236 \pm 0.0040$ | $0.1157 \pm 0.0081$ | $0.8047 \pm 0.0088$ |
| SCIPENN | 5 | $0.8305 \pm 0.0100$ | $0.8241 \pm 0.0080$ | $0.0037 \pm 0.0008$ | $0.1960 \pm 0.0122$ | $0.5538 \pm 0.0051$ |
| SCVAEIT | 5 | $0.8525 \pm 0.0045$ | $0.9295 \pm 0.0046$ | $0.2543 \pm 0.0281$ | $0.0758 \pm 0.0053$ | $0.7716 \pm 0.0082$ |
| TOTALVI | 5 | $0.6629 \pm 0.0321$ | $0.8014 \pm 0.0081$ | $0.1077 \pm 0.0106$ | $0.1491 \pm 0.0110$ | $0.9779 \pm 0.0513$ |
| **SCCHORD** | 5 | $\mathbf{0.8655 \pm 0.0045}$ | $\mathbf{0.9392 \pm 0.0013}$ | $\mathbf{0.0029 \pm 0.0004}$ | $\mathbf{0.0316 \pm 0.0011}$ | $\mathbf{0.5384 \pm 0.0045}$ |

*Table 7.* Performance comparison on the **GSE164378-3P** dataset. Results reflect mean $\pm$ standard deviation across $n = 5$ independent experiments.

| Method | PCC-P $\uparrow$ | PCC-C $\uparrow$ | CMD-P $\downarrow$ | CMD-C $\downarrow$ | RMSE $\downarrow$ |
|---|---|---|---|---|---|
| cTP-net | $0.4033 \pm 0.0125$ | $0.2884 \pm 0.0142$ | $0.6543 \pm 0.0210$ | $0.1424 \pm 0.0085$ | $1.0236 \pm 0.0284$ |
| Dengkw | $0.5171 \pm 0.0284$ | $0.8744 \pm 0.0062$ | $0.1791 \pm 0.0154$ | $0.1024 \pm 0.0118$ | $1.0655 \pm 0.0075$ |
| LIGER | $0.4628 \pm 0.0132$ | $0.8678 \pm 0.0041$ | $\mathbf{0.1049 \pm 0.0102}$ | $0.1009 \pm 0.0045$ | $1.0254 \pm 0.0112$ |
| scArches | $0.4400 \pm 0.0085$ | $0.8893 \pm 0.0114$ | $0.2725 \pm 0.0052$ | $0.0597 \pm 0.0092$ | $1.0405 \pm 0.0094$ |
| sciPENN | $\underline{0.5243 \pm 0.0112}$ | $0.5646 \pm 0.0095$ | $0.1678 \pm 0.0012$ | $0.1363 \pm 0.0135$ | $\underline{0.7840 \pm 0.0062}$ |
| scVAEIT | $0.5132 \pm 0.0051$ | $0.8887 \pm 0.0052$ | $0.4307 \pm 0.0315$ | $0.1466 \pm 0.0064$ | $1.0256 \pm 0.0093$ |
| totalVI | $0.5169 \pm 0.0354$ | $\mathbf{0.9243 \pm 0.0075}$ | $0.1722 \pm 0.0125$ | $\underline{0.0529 \pm 0.0128}$ | $1.0257 \pm 0.0582$ |
| **scChord** | $\mathbf{0.5274 \pm 0.0042}$ | $\underline{0.8934 \pm 0.0128}$ | $\underline{0.1690 \pm 0.0155}$ | $\mathbf{0.0033 \pm 0.0008}$ | $\mathbf{0.7779 \pm 0.0052}$ |

| oprule Variant | Stage 1 | Stage 2 | Encoder update | PCC-P $\uparrow$ | PCC-C $\uparrow$ | CMD-P $\downarrow$ | CMD-C $\downarrow$ | RMSE $\downarrow$ |
|---|---|---|---|---|---|---|---|---|
| Joint training | joint | joint | yes | 0.8655 | 0.9392 | 0.0029 | 0.0316 | 0.5384 |
| Pretrain→Finetune | pretrained | finetuned | yes | 0.8596 | 0.9428 | 0.0038 | 0.0310 | 0.5475 |
| Pretrain→Freeze | pretrained | frozen | no | 0.7889 | 0.8657 | 0.0045 | 0.0410 | 0.6018 |

**Results** Statistical tests indicated highly significant differences among methods for both metrics (Table **??**). For PCC-P, the large effect sizes ($\eta^2 = 0.607$, $W = 0.809$) suggest that method choice explains a substantial fraction of performance variance.

### C.2. Extended visualizations of protein distribution reconstruction

To comprehensively evaluate the distributional fidelity of scChord, we extended our analysis to a diverse set of 11 surface protein markers. These markers were categorized into three biologically distinct groups to assess performance across different distributional landscapes: T-cell markers (CD3, CD8), NK/B-cell and Myeloid markers (CD56, CD16, CD10, CD11c, CD14, CD19), and Stem/Chemokine receptors (CD34, CCR5, CCR7). We compared scChord against eight baselines using the Kolmogorov-Smirnov (KS) statistic and visual inspection of Kernel Density Estimation (KDE) curves.

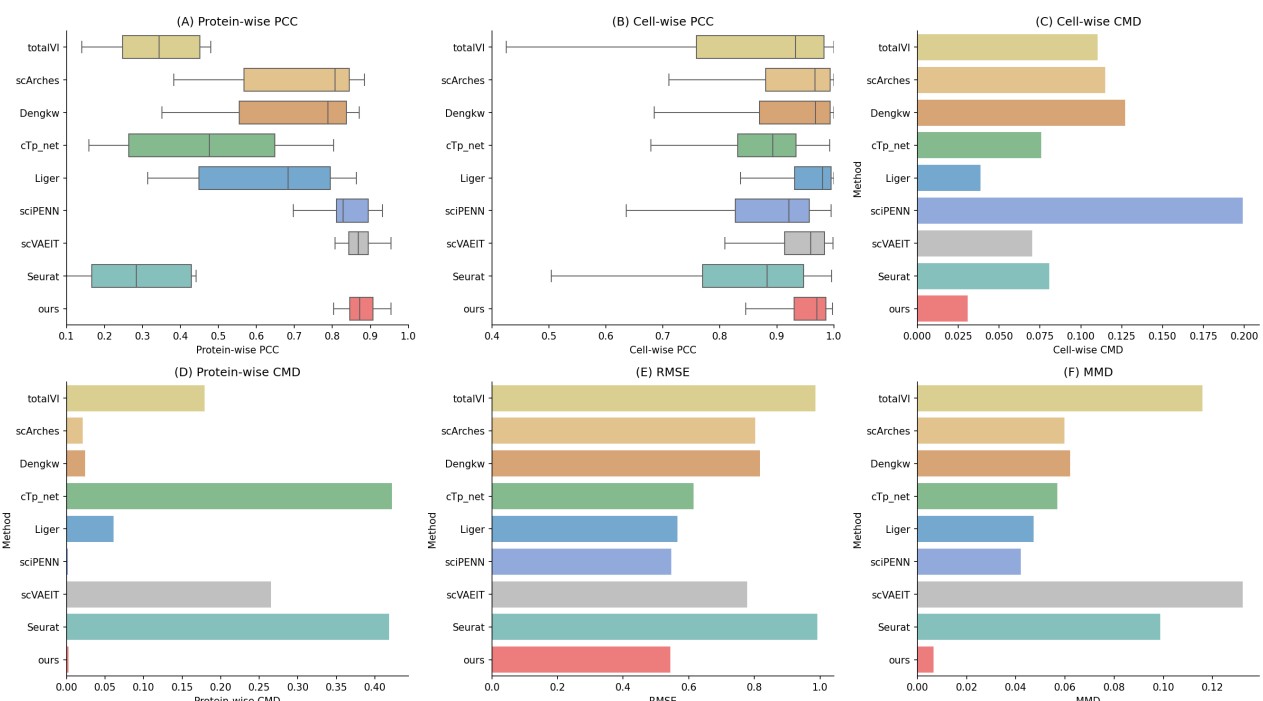

*Figure 8.* **Summary of evaluation metrics on GSE100866.** We report PCC at the protein level (13 proteins) and the cell level (1,723 cells) together with additional metrics.

**Recovery of Bimodal Populations (T-cell Markers).** Markers such as CD3, CD8, and the chemokine receptor CCR5 typically exhibit clear bimodal distributions, representing distinct negative and positive populations. As summarized in Table 8, `scChord` consistently captures this bimodality with high fidelity (e.g., CD3: 0.089, CD8: 0.084, CCR5: 0.093). In contrast, regression-based methods like Seurat often suffer from mode collapse, merging two distinct peaks into a single inaccurate mean (e.g., Seurat on CD3, KS=0.496), while `cTp-net` frequently introduces artificial oscillations.

**Mitigation of Zero-Inflation Artifacts (NK, B-cell, and Myeloid Markers).** This group (CD56, CD16, CD10, CD11c, CD14, CD19) presents a significant challenge due to data sparsity and complex subpopulations. A critical observation is the prevalence of *zero-inflation artifacts* in competing generative models. While methods like `scArches` and `Dengkw` achieve low KS scores on sparse markers (e.g., CD56, CD11c), visual inspection reveals sharp, non-biological spikes at zero expression. These artifacts suggest that these models incorrectly force low-expression cells to exact zeros to minimize statistical divergence. `scChord` avoids this pitfall, producing smooth, continuous tails that better reflect the biological reality of background noise and low-level expression (e.g., CD19 KS=0.113 vs. totalVI KS=0.327), maintaining the best overall average KS score (0.087) across all markers.

**Preservation of Continuous Spectrums (Stem Cell and Differentiation Markers).** For markers associated with continuous differentiation processes, such as CD34 (HSCs) and CCR7, the true distributions are often heavy-tailed or "smeared." `scChord` excels in these scenarios (CCR7 KS=0.068), accurately tracing the broad shoulders of the distribution where differentiation intermediates reside. Conversely, `totalVI` tends to shift the distribution means, and `sciPENN` fails to capture the coherent shape, resulting in noisy predictions that obscure the developmental trajectory.

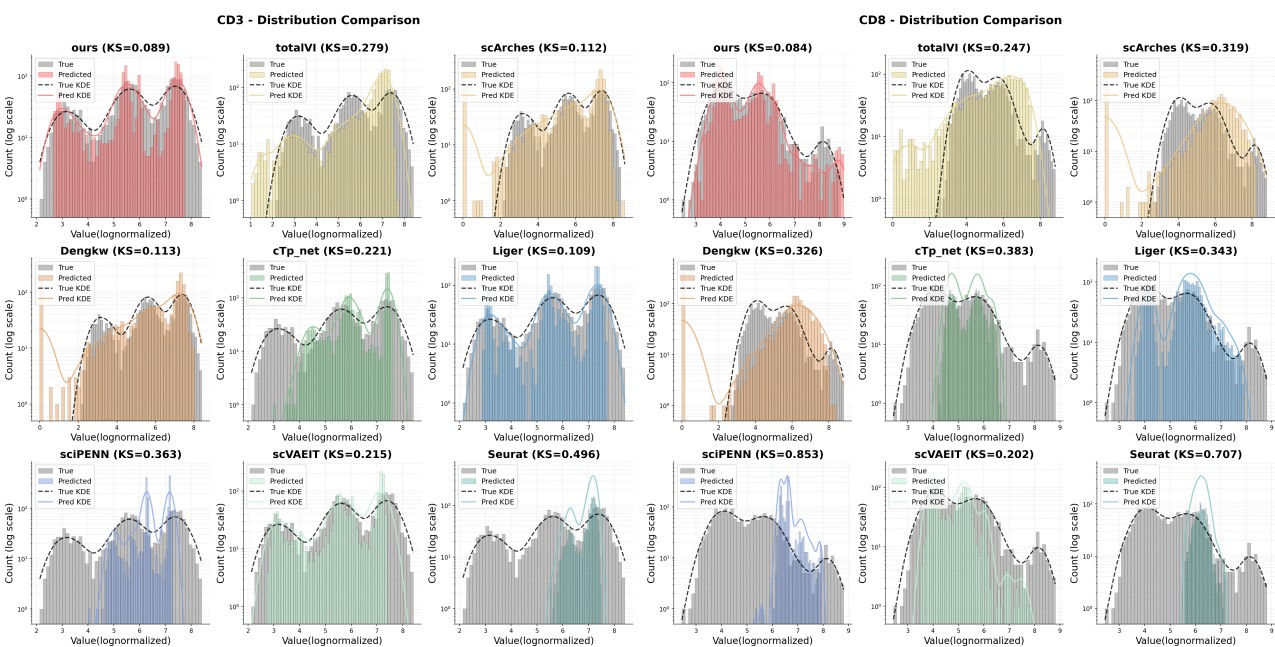

*Figure 9.* **Extended distribution reconstructions (T-cell markers).** CD3 and CD8 show characteristic bimodality. Visual inspection aligns with Table 8: scChord preserves multimodal structure while regression baselines (e.g., Seurat) often collapse modes.

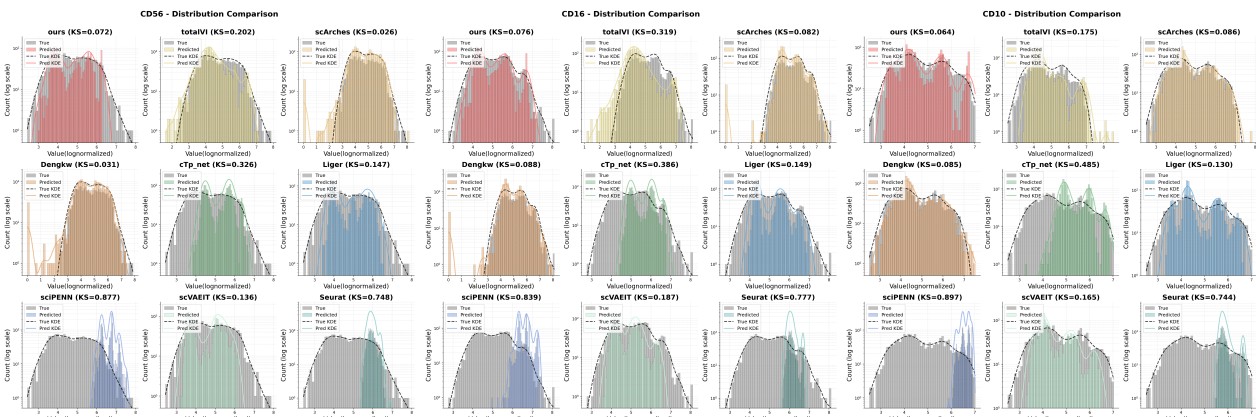

*Figure 10.* **Extended distribution reconstructions (sparse markers, part I).** CD56/CD16/CD10 are sensitive to sparsity-induced failure modes. Several generative baselines can match KS on some sparse markers yet introduce non-biological spikes at exactly zero; scChord yields smoother low-expression behavior.

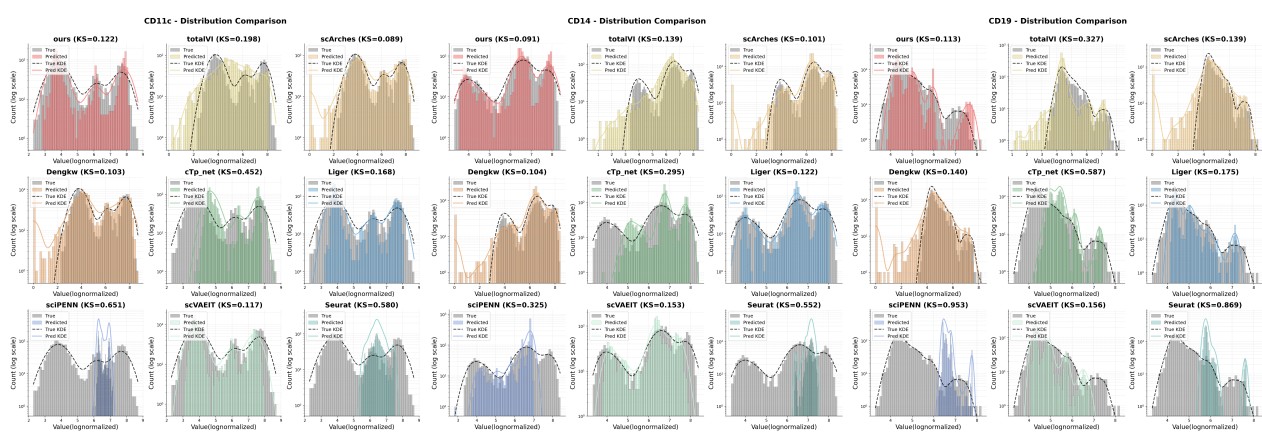

*Figure 11.* **Extended distribution reconstructions (sparse markers, part II).** CD11c/CD14/CD19 highlight that distributional fidelity is not only about matching central tendency but also about preserving tails and avoiding zero-inflation artifacts. scChord remains robust across markers.

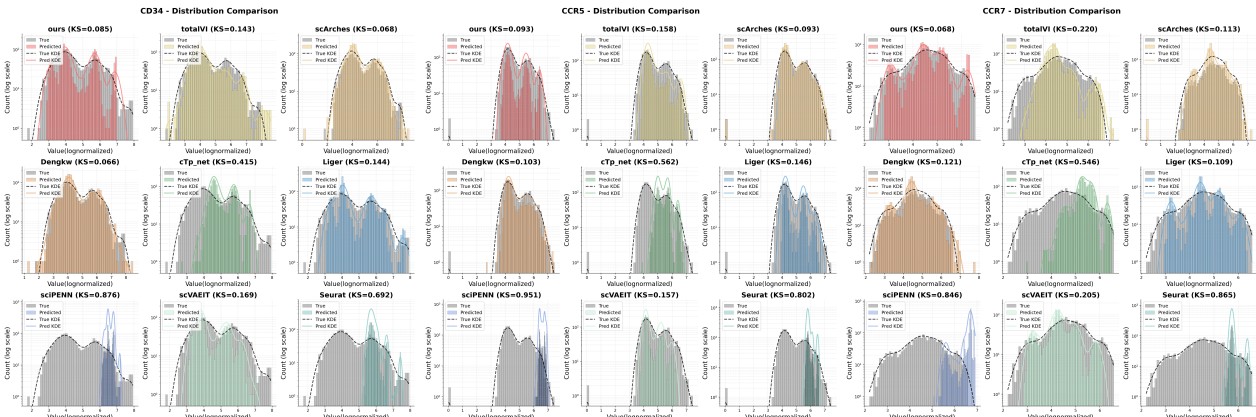

*Figure 12.* **Extended distribution reconstructions (Stem/Chemokine receptors).** CD34 exhibits a heavy-tailed continuum, while CCR5/CCR7 reflect chemokine-receptor-specific heterogeneous expression. scChord better follows broad shoulders and tails, supporting improved recovery of intermediate states.

*Table 8.* Comparison of Kolmogorov-Smirnov (KS) statistics across 11 protein markers on the GSE100866 dataset. Lower values indicate better distributional matching. scChord achieves the lowest average KS score and ranks first in 7 out of 11 markers.

| Method | CD3 | CD8 | CD56 | CD16 | CD10 | CD11c | CD14 | CD19 | CD34 | CCR5 | CCR7 | Avg KS |
|---|---|---|---|---|---|---|---|---|---|---|---|---|
| **scChord (Ours)** | **0.089** | **0.084** | 0.072 | **0.076** | **0.064** | 0.122 | **0.091** | **0.113** | 0.085 | **0.093** | **0.068** | **0.087** |
| scArches | 0.112 | 0.319 | **0.026** | 0.082 | 0.086 | **0.089** | 0.101 | 0.139 | 0.068 | **0.093** | 0.113 | 0.112 |
| Dengkw | 0.113 | 0.326 | 0.031 | 0.088 | 0.085 | 0.103 | 0.104 | 0.140 | **0.066** | 0.103 | 0.121 | 0.116 |
| Liger | 0.109 | 0.343 | 0.147 | 0.149 | 0.130 | 0.168 | 0.122 | 0.175 | 0.144 | 0.146 | 0.109 | 0.158 |
| scVAEIT | 0.215 | 0.202 | 0.136 | 0.187 | 0.165 | 0.117 | 0.153 | 0.156 | 0.169 | 0.157 | 0.205 | 0.169 |
| totalVI | 0.279 | 0.247 | 0.202 | 0.319 | 0.175 | 0.198 | 0.139 | 0.327 | 0.143 | 0.158 | 0.220 | 0.219 |
| cTp-net | 0.221 | 0.383 | 0.326 | 0.386 | 0.485 | 0.452 | 0.295 | 0.587 | 0.415 | 0.562 | 0.546 | 0.423 |
| Seurat | 0.496 | 0.707 | 0.748 | 0.777 | 0.744 | 0.580 | 0.552 | 0.869 | 0.692 | 0.802 | 0.865 | 0.684 |
| sciPENN | 0.363 | 0.853 | 0.877 | 0.839 | 0.897 | 0.651 | 0.325 | 0.953 | 0.876 | 0.951 | 0.846 | 0.766 |

*Note: While baselines like scArches achieve lower KS scores on specific sparse markers (e.g., CD56, CD11c), they exhibit significant*

*zero-inflation artifacts (spikes at 0) not present in the ground truth. scChord maintains the most consistent performance across all categories without such artifacts.*

## C.3. Statistical Properties and Stochastic Variability

To address whether distributional metrics alone are sufficient to characterize generation quality, we further quantified mean fidelity, variance preservation, and noise consistency on the GSE100866 test set. These statistics clarify an important failure mode of deterministic models: they can match the mean reasonably well while still collapsing variance and distorting stochastic noise structure.

As summarized in Table 9, scChord attains the strongest overall alignment across all four statistics. In particular, the gap in variance and noise correlation is more informative than mean correlation alone, because mean expression is relatively easy to approximate even when a model suppresses stochastic variability. Accordingly, these results support the view that scChord better preserves the intrinsic stochasticity of protein translation rather than only matching average expression levels.

*Table 9.* Statistical property comparison on the GSE100866 test set. Higher is better for correlations, while lower is better for Noise Median APE.

| METHOD | MEAN CORR ↑ | VARIANCE CORR ↑ | NOISE CORR ↑ | NOISE MEDIAN APE ↓ |
|---|---|---|---|---|
| SCCHORD (OURS) | **0.9995** | **0.9917** | **0.9903** | **0.0505** |
| SCVAEIT | 0.9989 | 0.9837 | 0.9773 | 0.3171 |
| LIGER | 0.9904 | 0.9878 | 0.9728 | 0.2416 |
| CTP-NET | 0.9921 | 0.9418 | 0.9464 | 0.6536 |
| TOTALVI | 0.9770 | 0.7716 | 0.7917 | 0.2298 |
| DENGKW | 0.9910 | 0.4014 | 0.4517 | 0.4561 |
| SCARCHES | 0.9901 | 0.3904 | 0.4143 | 0.3469 |
| SEURAT | 0.9649 | 0.7310 | 0.6947 | 0.9576 |

# D. Quantitative Geometric Evaluation

To thoroughly quantify the geometric improvements provided by Stage 1 probabilistic rectification, we performed an extended evaluation on the GSE100866 test set comparing the Rectified Manifold against the deterministic MSE baseline across several nonlinear structural metrics.

As shown in Table 10, the rectified manifold consistently outperforms the unrectified baseline across all nonlinear geometric metrics. Notably, the larger Laplacian spectral gap and the 39% improvement in $R^2$ for intrinsic coordinates (0.407 vs. 0.292) confirm a more globally coherent topology that aligns better with intrinsic biological signals. Furthermore, the 26% reduction in the Local Intrinsic Dimension (ID) standard deviation indicates that rectification successfully mitigates localized overfitting to stochastic noise, directly supporting the theoretical diagnosis regarding Lipschitz bounds in Proposition A.4 (see Appendix A.1). While linear dimensionality (PCA PC@90%) remains identical between the two manifolds, the improvements in diffusion embeddings, eigenspace stability, and Dirichlet energy provide strong evidence that Stage 1 serves as an effective geometric preconditioner.

# E. Algorithmic Description

*Table 10.* **Quantitative geometric analysis on the GSE100866 test set.** We compare the properties of the learned latent manifold with (Rectified) and without (Original) probabilistic Stage 1. ↑ indicates higher is better, ↓ indicates lower is better.

| METRIC | RECTIFIED (SCCHORD) | ORIGINAL BASELINE |
|---|---|---|
| LAPLACIAN SPECTRAL GAP ↑ | **0.002274** | 0.0000449 |
| DIFFMAP MEAN $R^2$ ↑ | **0.407** | 0.292 |
| DIFFMAP DIST SPEARMAN ↑ | **0.596** | 0.514 |
| EIGENMAP MEAN $R^2$ ↑ | **0.407** | 0.292 |
| EIGENMAP DIST SPEARMAN ↑ | **0.596** | 0.514 |
| TWONN GLOBAL ID ↓ | **3.630** | 4.008 |
| TWONN LOCAL ID STD ↓ | **0.571** | 0.774 |
| DIRICHLET ENERGY ↓ | **0.1080** | 0.1091 |
| PCA PC@90% ↓ | 4 | 4 |

---

**Algorithm 1** Stage 1: ProteinVAE Training

---

**Require:** Protein measurements $\{(\mathbf{y}_i^{\text{prot}}, \mathbf{y}_i^{\text{prot,raw}}, b_i)\}_{i=1}^{N}$, where $\mathbf{y}^{\text{prot}} \in \mathbb{R}^M$ is normalized protein abundance, $\mathbf{y}^{\text{prot,raw}}$ are raw counts, and $b \in \{1, \ldots, B\}$ is the batch ID.
1: Initialize VAE parameters $\phi = \{\phi_{\text{enc}}, \phi_{\text{dec}}\}$.
2: Initialize learnable dispersion $\boldsymbol{\theta} \in \mathbb{R}^M$ (gene-wise / protein-wise).
3: Initialize dropout head $\pi_{\text{decoder}}$ (cell-wise prediction) for ZINB likelihood.
4: **for** each epoch **do**
5:     **for** each mini-batch $(\mathbf{y}^{\text{prot}}, \mathbf{y}^{\text{prot,raw}}, \mathbf{b})$ **do**
6:         **Encoder (posterior parameters)**
7:         $\mathbf{h} = \text{Encoder}([\mathbf{y}^{\text{prot}}; \text{Embed}(\mathbf{b})])$
8:         $\boldsymbol{\mu}_z, \log \boldsymbol{\sigma}_z^2 = \text{FC}_\mu(\mathbf{h}), \text{FC}_{\log \sigma^2}(\mathbf{h})$
9:         $\mathbf{z} = \boldsymbol{\mu}_z + \boldsymbol{\sigma}_z \odot \boldsymbol{\epsilon}, \quad \boldsymbol{\epsilon} \sim \mathcal{N}(\mathbf{0}, \mathbf{I})$         // reparameterization
10:       **Decoder (ZINB parameters)**
11:       $\hat{\boldsymbol{\mu}} = \text{Decoder}([\mathbf{z}; \text{Embed}(\mathbf{b})])$         // mean (cell-wise)
12:       $\boldsymbol{\pi}_{\text{logit}} = \pi_{\text{decoder}}(\mathbf{h}_{\text{intermediate}})$         // dropout logits
13:       **Loss**
14:       $\mathcal{L}_{\text{recon}} = -\sum_m \log \text{ZINB}(y_m^{\text{raw}} | \hat{\mu}_m, \theta_m, \pi_m)$
15:       $\mathcal{L}_{\text{KL}} = \frac{1}{2} \sum_j \left[ \mu_{z,j}^2 + \sigma_{z,j}^2 - 1 - \log \sigma_{z,j}^2 \right]$
16:       $\mathcal{L} = \mathcal{L}_{\text{recon}} + \beta_{\text{KL}} \cdot \mathcal{L}_{\text{KL}}$
17:       Update $\phi, \boldsymbol{\theta}$, and $\pi_{\text{decoder}}$.
18:     **end for**
19: **end for**
20: **Output:** Trained VAE (encoder and decoder).

---

---

**Algorithm 2** Stage 2: Conditional Flow Matching Training

---

**Require:** Paired data $\{(\mathbf{x}_i^{\mathrm{rna}}, \mathbf{y}_i^{\mathrm{prot}}, b_i)\}_{i=1}^N$.
**Require:** Pretrained and frozen VAE from Stage 1.
**Require:** Hyperparameters: $p_{\mathrm{uncond}}$ (unconditional probability), $\lambda_{\mathrm{cons}}$ (consistency weight), mask ratio range $(r_{\min}, r_{\max})$.
1: Initialize RNAEncoder parameters $\psi$.
2: Initialize FlowNet parameters $\theta$ (with learnable unconditional embedding $\mathbf{c}_\varnothing \in \mathbb{R}^{d_c}$).
3: **for** each epoch **do**
4:    **for** each mini-batch $(\mathbf{x}^{\mathrm{rna}}, \mathbf{y}^{\mathrm{prot}}, \mathbf{b})$ **do**
5:       **Step 1: target endpoint sampling**
6:       $\boldsymbol{\mu}_z, \log \boldsymbol{\sigma}_z^2 = \mathrm{VAE.Encode}(\mathbf{y}^{\mathrm{prot}}, \mathbf{b})$      // posterior params
7:       $\mathbf{x}_1 = \boldsymbol{\mu}_z + \boldsymbol{\sigma}_z \odot \boldsymbol{\epsilon}, \quad \boldsymbol{\epsilon} \sim \mathcal{N}(\mathbf{0}, \mathbf{I})$      // posterior sampling
8:       **Step 2: conditioning + consistency**
9:       $\mathbf{c}_{\mathrm{full}} = \mathrm{RNAEncoder}(\mathbf{x}^{\mathrm{rna}}, \mathbf{b})$      // full condition
10:      // **Masking**: sample expressed genes only
11:      $r \sim \mathrm{Uniform}(r_{\min}, r_{\max})$      // sample mask ratio
12:      $S_i \subseteq \{g : x_{ig}^{\mathrm{rna}} > 0\}, |S_i| = \lfloor r \cdot |\{g : x_{ig}^{\mathrm{rna}} > 0\}| \rfloor$
13:      $\tilde{x}_{ig}^{\mathrm{rna}} = 0$ for $g \in S_i$, and $\tilde{x}_{ig}^{\mathrm{rna}} = x_{ig}^{\mathrm{rna}}$ otherwise      // mask expressed entries only
14:      $\mathbf{c}_{\mathrm{mask}} = \mathrm{RNAEncoder}(\tilde{\mathbf{x}}^{\mathrm{rna}}, \mathbf{b})$
15:      $\mathcal{L}_{\mathrm{cons}} = \|\mathbf{c}_{\mathrm{full}} - \mathbf{c}_{\mathrm{mask}}\|_2^2$      // consistency
16:      **Step 3: CFG dropout**
17:      $u \sim \mathrm{Bernoulli}(p_{\mathrm{uncond}})$      // sample CFG dropout flag
18:      $\mathbf{c}_{\mathrm{used}} = \begin{cases} \mathbf{c}_\varnothing & \text{if } u = 1 \\ \mathbf{c}_{\mathrm{full}} & \text{if } u = 0 \end{cases}$
19:      **Step 4: flow matching**
20:      $\mathbf{x}_0 \sim \mathcal{N}(\mathbf{0}, \mathbf{I})$      // source noise
21:      $t \sim \mathrm{Uniform}(0, 1)$      // time
22:      $\mathbf{x}_t = (1 - t)\mathbf{x}_0 + t\mathbf{x}_1$      // linear path
23:      $\mathbf{u}_t = \mathbf{x}_1 - \mathbf{x}_0$      // target vector field
24:      $\mathbf{v}_\theta = \mathrm{FlowNet}(\mathbf{x}_t, t, \mathbf{c}_{\mathrm{used}}, \mathbf{b})$      // conditioned via AdaLN
25:      $\mathcal{L}_{\mathrm{CFM}} = \|\mathbf{v}_\theta - \mathbf{u}_t\|_2^2$      // CFM loss
26:      $\mathcal{L} = \mathcal{L}_{\mathrm{CFM}} + \lambda_{\mathrm{cons}} \cdot \mathcal{L}_{\mathrm{cons}}$
27:      Update $\psi, \theta$.
28:    **end for**
29: **end for**
30: **Output:** Trained RNAEncoder and FlowNet.

---

**Algorithm 3** Inference: RNA to Protein Prediction

---

**Require:** RNA expression $\mathbf{x}^{\mathrm{rna}}$ and batch ID $b$.
**Require:** Trained RNAEncoder, FlowNet, and VAE decoder.
**Require:** CFG weight $w > 1$.
1: $\mathbf{c} = \mathrm{RNAEncoder}(\mathbf{x}^{\mathrm{rna}}, b)$      // condition embedding
2: $\mathbf{x}_0 \sim \mathcal{N}(\mathbf{0}, \mathbf{I})$      // initial noise
3: **ODE integration (with CFG)**
4: **for** $t$ from 0 to 1 (using an ODE solver, e.g., Dopri5) **do**
5:    $\mathbf{v}_{\mathrm{cond}} = \mathrm{FlowNet}(\mathbf{x}_t, t, \mathbf{c}, b)$
6:    $\mathbf{v}_{\mathrm{uncond}} = \mathrm{FlowNet}(\mathbf{x}_t, t, \mathbf{c}_\varnothing, b)$
7:    $\mathbf{v} = \mathbf{v}_{\mathrm{uncond}} + w \cdot (\mathbf{v}_{\mathrm{cond}} - \mathbf{v}_{\mathrm{uncond}})$      // CFG
8:    $\mathbf{x}_{t+\Delta t} = \mathrm{ODEStep}(\mathbf{x}_t, \mathbf{v}, \Delta t)$
9: **end for**
10: $\hat{\mathbf{z}} = \mathbf{x}_1$      // final latent
11: $\hat{\mathbf{y}}^{\mathrm{prot}} = \mathrm{VAE.Decode}(\hat{\mathbf{z}}, b)$      // decode protein
12: **Output:** $\hat{\mathbf{y}}^{\mathrm{prot}}$.

---

