# OpenReview forum: "scChord: A Probabilistic Manifold Rectification Framework for RNA-to-Protein Translation"
_ICML.cc/2026/Conference — ICML 2026 regular_

### Official Review · Reviewer_zCLM · 2026-03-09

**Soundness:** 3
**Presentation:** 2
**Significance:** 3
**Originality:** 2
**Overall Recommendation:** 4
**Confidence:** 3

**Summary:**

The paper proposes scChord, a probabilistic framework for translating single-cell RNA-seq data to protein abundance. It employs a two-stage approach: first, a probabilistic manifold rectification module learns a smoothed protein latent representation using a ZINB VAE; second, a consistency-regularized conditional flow matching model maps RNA embeddings to this rectified space. Extensive experiments on multiple CITE-seq datasets demonstrate that scChord outperforms existing methods on standard metrics.

**Compliance With Llm Reviewing Policy:**

Affirmed.

**Final Justification:**

I thank the authors for their response. As most of my concerns have been addressed, I assign a score of 4.

**Key Questions For Authors:**

1.	Could the authors further analyze the statistical properties of the predicted protein distributions—such as mean–variance relationships—in comparison with the observed distributions?

2.	If the improvement in CMD is interpreted as capturing gene regulatory relationships, could the authors validate this claim using known regulatory networks or synthetic datasets with known regulatory structures?

3.	Could the authors report computational efficiency on large-scale datasets?

4.	Evaluate whether the model captures stochastic variability (e.g., variance/mean²) in Figure 7, rather than only reporting distribution-level fitting results.

5.	While the strong predictive performance of scChord on the GSE164378-3P dataset is clearly demonstrated in Table 2, claims regarding scalability would be further strengthened by including quantitative efficiency metrics. We encourage the authors to report training time, GPU memory usage, and an overall efficiency comparison with baseline methods alongside the existing accuracy results.

**Limitations:**

yes

**Strengths And Weaknesses:**

**Strengths**

1.	Motivation: The work tackles a timely and practically relevant problem in single-cell biology, aligning well with current experimental and analytical needs in multi-omics integration.

2.	Methodology: The two-stage framework is clearly structured and logically motivated, combining probabilistic manifold rectification with conditional flow matching in an interpretable manner.

3.	Experiments: Comprehensive benchmarking across multiple datasets, along with ablation studies and robustness analyses, provides strong empirical support for the method's effectiveness and generalizability.

**Weaknesses**

1.	The improvement in CMD is interpreted as capturing gene regulatory relationships, but CMD mainly measures correlation structure and does not necessarily imply regulatory or causal relationships.

2.	This paper emphasizes the model's applicability to large-scale datasets, but does not report computational metrics such as training time, memory usage, or inference time.

---

> ### Author Rebuttal · Authors · 2026-03-31
>
> We sincerely thank you for your rigorous evaluation and for recognizing the timely motivation, structured methodology, and comprehensive benchmarking of our work. Your insightful comments regarding the statistical interpretation of our metrics and the need for computational efficiency reporting have significantly strengthened the manuscript.
>
> Below is our point-by-point response to your weaknesses and key questions.
>
> ### 1. Interpretation of CMD (W1, Q2)
> *(Addressing Weakness 1 & Key Question 2)*
>
> We completely agree that CMD measures statistical correlation rather than direct causal or regulatory mechanisms. Our original phrasing was imprecise. We intended to emphasize that scChord captures high-order co-expression patterns and statistical dependencies between transcripts and proteins. To ensure absolute rigor, we have thoroughly revised the manuscript (Sections 4.1 and 5), replacing all instances of "regulatory relationships/networks" with precise statistical terminology, such as "complex co-expression patterns" and "intrinsic correlation structures."
>
> ### 2. Statistical Properties and Stochastic Variability
> *(Addressing Key Questions 1 & 4)*
>
> This constructive suggestion highlights our probabilistic framework's advantage over deterministic models, which inherently collapse variance. Following your recommendation, we computed the Squared Coefficient of Variation ($CV^2 = \mathrm{Variance}/\mathrm{Mean}^2$) to quantify stochastic noise and calculated the Noise Median Absolute Percentage Error (APE). These results are now in the Appendix:
>
> | Method            | Mean Corr | Variance Corr | Noise Corr ($CV^2$) | Noise Median APE |
> |-------------------|-----------|---------------|---------------------|------------------|
> | **scChord (Ours)**| **0.9995**| **0.9917**    | **0.9903**          | **0.0505**       |
> | scVAEIT           | 0.9989    | 0.9837        | 0.9773              | 0.3171           |
> | Liger             | 0.9904    | 0.9878        | 0.9728              | 0.2416           |
> | cTp_net           | 0.9921    | 0.9418        | 0.9464              | 0.6536           |
> | totalVI           | 0.9770    | 0.7716        | 0.7917              | 0.2298           |
> | Dengkw            | 0.9910    | 0.4014        | 0.4517              | 0.4561           |
> | scArches          | 0.9901    | 0.3904        | 0.4143              | 0.3469           |
> | Seurat            | 0.9649    | 0.7310        | 0.6947              | 0.9576           |
>
> While most methods capture mean expression, deterministic approaches (e.g., Seurat, cTp_net, Dengkw) show a severe drop in Variance/Noise Correlation and high APEs, confirming variance collapse. Conversely, scChord almost perfectly preserves intrinsic stochastic variability (Noise Corr=0.9903, lowest APE=0.0505), validating that our rectified manifold effectively models the stochastic nature of protein translation.
>
> ### 3. Computational Efficiency and Scalability
> *(Addressing Weakness 2 & Key Questions 3 & 5)*
>
> We thank the reviewer for raising this practical concern. We benchmarked all models at scales of 1k, 10k, 50k, and 100k cells (GSE164378-3P, NVIDIA RTX 4080); full scaling curves are in the revised version and in our anonymous repository linked at the end of the paper.
>
> | Method | Train (s) | CPU Mem (MB) | GPU Mem (MB) | Infer (s/200) |
> |---|---|---|---|---|
> | cTp_net | — | — | — | 0.259 |
> | sciPENN | — | — | — | 0.002 |
> | Dengkw | 195 | 4,910 | 0 | 0.023 |
> | Liger | 403 | 36,356 | 0 | 0.026 |
> | scArches | 1,419 | 23,779 | 833 | 0.069 |
> | scVAEIT | 2,923 | 9,968 | 712 | 0.465 |
> | totalVI | 1,297 | 22,140 | 392 | 0.030 |
> | scChord | 1,636 | 10,766 | 229 | 0.488 |
> *(scChord total = Stage 1: 576s + Stage 2: 1,061s.)*
> **Note:** For cTp_net and sciPENN, training and memory metrics are unavailable because both methods crashed with out-of-memory errors (exceeding 64GB RAM) at the 100k cell scale.
>
> We would like to offer an honest and balanced assessment: (1) Training: Non-deep-learning methods (Dengkw, Liger) train faster, which is expected given their lower modeling capacity. Among deep generative models, scChord is competitive—1.8× faster than scVAEIT and comparable to totalVI. (2) Inference: We acknowledge scChord's inference latency is the highest, a genuine limitation inherent to multi-step ODE integration. We are actively pursuing two mitigation strategies: reducing integration steps and replacing dopri5 with a more efficient fixed-step solver (e.g., Euler or midpoint), both yielding substantial speedups with marginal accuracy loss in preliminary tests. (3) Memory: scChord achieves the lowest GPU peak memory (229 MB) among GPU-based methods and linear CPU scaling (~10.7 GB at 100k cells), far below Liger (36 GB) and totalVI (22 GB).
>
> We believe these additions thoroughly address your concerns and provide a transparent view of the model's practical utility.

---

> > ### Author Rebuttal · Reviewer_zCLM · 2026-04-02
> >
> > I thank the authors for their response. As most of my concerns have been addressed, I assign a score of 4.

---

> > > ### Author Response · Authors · 2026-04-03
> > >
> > > We sincerely thank the reviewer for the careful evaluation and for raising the score. We are truly grateful for the constructive feedback throughout this process, which has helped us meaningfully improve the work. We will incorporate all the suggested improvements, including computational efficiency metrics and the statistical property analyses, in the revised manuscript. Should the reviewer have any further questions or wish to discuss any aspect of the work, we would be very happy to continue the conversation!

---

### Official Review · Reviewer_T1rw · 2026-03-11

**Soundness:** 4
**Presentation:** 3
**Significance:** 3
**Originality:** 2
**Overall Recommendation:** 4
**Confidence:** 4

**Summary:**

This paper proposes a method for translating scRNAseq data to single cell protein count data.

The method consists of two stages:

In the first stage, it proposes to use a VAE with ZINB likelihood and interprets it as disentangling the uncertainty from the latent manifold. It provides theory in gradient scales and sensitivity for the interpretation.

In the second stage, an OT-CFM model is trained to generate the target data from noise. A noise-robustness regularization term is added to the flow matching training loss.

The model is evaluated on real-world benchmarks against a large number of baselines. Ablation studies also showed the effectiveness of the components.

**Compliance With Llm Reviewing Policy:**

Affirmed.

**Final Justification:**

During the rebuttal, the authors have addressed my concerns in terms of additional metrics, ablation study on design choice, and methodological novelty and motivation. Therefore, I have raised my score.

**Key Questions For Authors:**

1. Are there more direct ways to quantify the manifold landscape than interpolation, NFE, sensibility? Can you try to quantify the geometric properties of the manifold itself, such as using dimensionality reduction and some graph-based smoothness metrics (e.g. DiffusionMaps and Laplacian Eigenmap)? What about intrinsic dimension and PCA?
2. The distribution-level evaluation using KS Distance is good, but can you make a more systematic and quantitative comparison in a table? Some other common distribution-level metrics including MMD and EMD would be a good fit in addition to KS Distance.
3. Can you explain why train the RNA encoder jointly with the flow matching? That would add training cost and complicate the dynamics and induce a tradeoff of the two loss terms, compared to training the encoder first. Was this choice supported in any ablation?

**Limitations:**

Yes

**Strengths And Weaknesses:**

**Soundness**

The paper is technically sound, with rigorous theory, proofs, and abundant experiments. The experiments are repeated with statistical significance presented, and the ablations studies support the framework. The leave-group-out experiments are well-designed to show the model’s ability to generalize.

**Presentation**

The paper is generally well-written, but can be improved by making the problem formulation more explicit, such as adding a section defining the notations, and adding some clarifications on the data shape, etc. For example, it would benefit from clarifying that the source and target are both count data, instead of sequences and structures.

The paper has provided clear algorithm boxes, experiment details, and source code that are beneficial for reproducing the results.

**Significance**

The problem it tries to solve, RNA count to protein count translation is in general important and has been studied by a number of papers (as shown in related work and baselines).

**Originality**

This is where my largest concern of this paper lies. The two main methodological contributions seem to be 1) a VAE with ZINB likelihood for encoding proteins, and 2) a robustness regularization for training the RNA encoder by adding dropout noise and requiring the encoder to embed the noised data close to the original data. Both techniques are standard and widely used. Even though the authors provided theoretical interpretations for those techniques, this would note add to the methodological novelty.

---

> ### Author Rebuttal · Authors · 2026-03-31
>
> We sincerely thank the reviewer for the thorough and constructive feedback. We greatly appreciate your recognition of the soundness and experimental rigor of our work. Below, we carefully address each concern.
>
> **On Originality**
>
> scChord introduces a unified theoretical and architectural framework addressing a previously unarticulated geometric failure mode. Unlike methods (totalVI, scVAEIT, scDM) facing empirical instability, we prove (Prop. A.4) that deterministic constraints on heteroscedastic single-cell data mathematically necessitate high encoder sensitivity, causing uncertainty collapse and unstable ODE trajectories.
>
> To resolve this, scChord repurposes the ZINB decoder from a generative endpoint to a geometric preconditioner that absorbs heteroscedastic noise. This ensures the posterior mean $\mu_z$ resides on a smooth manifold (Appendix A.2 details how ZINB gradient scaling implicitly bounds the encoder's local Lipschitz constant). This is reinforced by $L_{cons}$, which specifically stabilizes the conditional vector field $v_\theta(z_t, t, c)$ against RNA dropout perturbations for robust ODE integration.
>
> **Q1: More direct quantification of manifold geometry.**
>
> We thank the reviewer for this valuable suggestion and have implemented all requested analyses on the GSE100866 test set, comparing the rectified manifold against the deterministic MSE baseline:
>
> | Metric | Rectified | Original |
> | :--- | :--- | :--- |
> | Laplacian Spectral Gap ↑ | **0.002274** | 0.0000449 |
> | DiffMap Mean R² ↑ | **0.407** | 0.292 |
> | DiffMap Dist Spearman ↑ | **0.596** | 0.514 |
> | EigenMap Mean R² ↑ | **0.407** | 0.292 |
> | EigenMap Dist Spearman ↑ | **0.596** | 0.514 |
> | TwoNN Global ID ↓ | **3.630** | 4.008 |
> | TwoNN Local ID Std ↓ | **0.571** | 0.774 |
> | Dirichlet Energy ↓ | **0.1080** | 0.1091 |
> | PCA PC@90% ↓ | 4 | 4 |
>
> The rectified manifold consistently outperforms the baseline across all nonlinear geometric metrics. Notably, the larger Laplacian spectral gap and the 39% improvement in R² for intrinsic coordinates (0.407 vs. 0.292) confirm a more globally coherent topology that aligns better with biological signals. Furthermore, the 26% reduction in Local ID standard deviation indicates that rectification successfully mitigates localized overfitting to stochastic noise, directly supporting the theoretical diagnosis in Proposition A.4.
>
> While linear global variance (PCA PC@90%) remains identical, the significant gains in nonlinear metrics demonstrate that scChord provides essential localized geometric regularization. We will incorporate this quantitative analysis into Section 4.2 of the revised manuscript.
>
> **Q2: Systematic distribution-level evaluation with MMD and EMD.**
>
> We thank the reviewer for this helpful suggestion. Due to space constraints in the main text, the MMD results are currently presented in Appendix Figure 8(f), where scChord achieves the top performance on the GSE100866 dataset. To improve the accessibility of these comparisons, we will incorporate a consolidated table in the revised manuscript.
>
> **Q3: Why train the RNA encoder jointly with flow matching?**
>
> We thank the reviewer for raising this key design question. Any RNA-only pre-training objective optimizes for *intra-modal* structure with no supervision from the protein modality, creating a semantic misalignment with the flow matching target space. In fact, during the early stages of our work, we explored replacing the jointly-trained RNA encoder with frozen representations from scGPT — a large-scale RNA foundation model pre-trained on tens of millions of cells:
>
> | | PCC-P ↑ | PCC-C ↑ | CMD-P ↓ | CMD-C ↓ | RMSE ↓ |
> | :--- | :--- | :--- | :--- | :--- | :--- |
> | scChord (joint) | **0.8655** | **0.9392** | **0.0029** | **0.0316** | **0.5384** |
> | scGPT (frozen) | 0.7992 | 0.9009 | 0.0146 | 0.0547 | 0.6656 |
>
> Performance drops consistently across all metrics confirming that task-agnostic pre-training cannot substitute for task-aware joint optimization. This is mechanistically expected: $L_{cons}$ derives its effectiveness from being co-optimized with $L_{CFM}$, constraining the encoder to produce dropout-invariant embeddings *in the space that guides transport* — a coupling that a frozen encoder cannot provide. The stability of joint training is further supported by ablation Variant G (Table 3) and the low variance across five independent runs (Table 5). We will include this comparison in the revised manuscript.
>
> *All newly added experimental code and results has been open-sourced in the same anonymous repository linked at the end of the article.*
>
> **On Presentation.**
>
> We will add a dedicated notation section at the beginning of Section 3, explicitly clarifying that both source and target are count matrices from simultaneous multi-omics profiling, and will state data shapes explicitly at the start of Section 4.

---

> > ### Author Rebuttal · Reviewer_T1rw · 2026-04-01
> >
> > Thank you for the rebuttal!
> >
> > My concerns have been partially resolved by Q1 and Q2.
> > In Q3, even though the model outperforms scGPT, it does not directly show if joint training is better than pretrained-then-frozen, because the data, model architecture, and training setup between this model and scGPT are completely different. An ablation study controlling the same model, data would be ideal.
> >
> > Regarding originality, I understand that the paper provides a new theoretical perspective in terms of sensitivity and connects the optimization process with manifold smoothness. However, they do not directly motivate a new computational method, but are rather explanations/interpretations of a method that is a combination of well-established methods, and therefore have limited novelty.

---

> > > ### Author Response · Authors · 2026-04-02
> > >
> > > **We sincerely thank the reviewer for the continued and thoughtful engagement.**
> > >
> > > ---
> > >
> > > **On Q3.**
> > >
> > > We fully agree that the scGPT comparison does not constitute a controlled ablation. Our initial motivation for using scGPT was the prior assumption that a foundation model trained on tens of millions of cells would provide stronger embeddings — however, we acknowledge this does not control for architecture and training setup. Following the reviewer's suggestion, we first conducted a controlled experiment using the identical architecture and dataset, pre-training the RNA encoder by regressing onto Stage 1 protein posterior means and then freezing it during CFM training (Pretrain→Freeze). This already confirms our original hypothesis: decoupling the consistency loss from the flow matching objective leads to consistent degradation across all metrics. Further inspired by the reviewer's pretraining idea, we additionally evaluated a fine-tuned variant (Pretrain→Finetune), where the pre-trained encoder is subsequently optimized jointly with CFM training:
> > >
> > > | | PCC-P ↑ | PCC-C ↑ | CMD-P ↓ | CMD-C ↓ | RMSE ↓ |
> > > |---|---|---|---|---|---|
> > > | scChord (joint) | **0.8655** | 0.9392 | **0.0029** | 0.0316 | **0.5384** |
> > > | Pretrain→Finetune | 0.8596 | **0.9428** | 0.0038 | **0.0310** | 0.5475 |
> > > | Pretrain→Freeze | 0.7889 | 0.8657 | 0.0045 | 0.0410 | 0.6018 |
> > >
> > > The Pretrain→Finetune variant proves highly competitive, surpassing joint training on PCC-C and CMD-C without any dedicated hyperparameter tuning, suggesting explicit cross-modal pre-training followed by fine-tuning as a promising direction for further improvement. We are genuinely grateful for the reviewer's insight and will include this analysis in the revised manuscript.
> > >
> > > ---
> > >
> > > **On Originality.**
> > >
> > > We would like to offer additional context on how scChord's design emerged, which we hope addresses the reviewer's concern more directly.
> > >
> > > Our starting observation was that VAE-based generative methods (scVAEIT, totalVI) consistently outperformed regression baselines, suggesting that explicit distribution modeling is beneficial for this task. This motivated us to explore flow matching and diffusion-based approaches, which have demonstrated strong performance in other generative modeling domains. However, as shown in Variant B of our ablation (Table 3), naively applying these methods yielded catastrophic results — despite the target space here (10–300 dimensional protein counts) being far lower-dimensional than typical image generation tasks.
> > >
> > > This failure demanded an explanation. Unlike image data, which does not suffer from technical dropouts, over-dispersion, or zero-inflation, single-cell measurements are subject to multi-factor stochasticity that substantially elevates apparent noise levels. Through theoretical analysis, we identified that this causes the latent manifold to become geometrically rough and high-curvature, making stable ODE integration intractable regardless of model capacity — a failure mode not previously identified in the literature.
> > >
> > > This diagnosis directly motivated the ZINB VAE as a geometric preconditioner: the theoretical framework was not developed to explain an arbitrary design choice, but to formalize the causal understanding of why Variant B failed and what structural property the solution must possess. We believe this distinction — between post-hoc interpretation and theory-driven diagnosis — is meaningful, and that the identified principle of manifold rectification as a prerequisite for continuous transport in noisy biological settings offers genuine guidance for future work. The subsequent addition of $\mathcal{L}_{cons}$ followed the same logic: the complementary challenge of RNA sparsity in the conditioning signal was independently identified, and the consistency regularization was introduced as a targeted solution to stabilize the conditional vector field against dropout-induced perturbations. We respectfully hope the reviewer may consider this perspective in the final evaluation.

---

### Official Review · Reviewer_Fv96 · 2026-03-12

**Soundness:** 4
**Presentation:** 3
**Significance:** 3
**Originality:** 3
**Overall Recommendation:** 5
**Confidence:** 4

**Summary:**

The paper proposes scChord, a novel two-stage generative framework designed for single-cell RNA-to-protein translation. The authors identify a fundamental geometric obstruction in existing methods: enforcing deterministic constraints on highly stochastic and heteroscedastic single-cell data causes "uncertainty collapse," resulting in a rough latent manifold that severely destabilizes continuous dynamics (like flow matching). To resolve this, scChord introduces a Probabilistic Manifold Rectification stage (using a VAE with a ZINB decoder) that absorbs technical noise and over-dispersion into distributional parameters, thereby smoothing the latent geometry. Subsequently, a Consistency-Regularized Flow Matching (CFM) stage learns conditional transport trajectories on this rectified manifold, using a transcriptomic consistency loss to handle RNA sparsity. Evaluated on multiple CITE-seq and ECCITE-seq benchmarks, scChord achieves state-of-the-art accuracy and demonstrates an exceptional ability to recover complex biological distributions, such as bimodal and heavy-tailed protein populations.

**Compliance With Llm Reviewing Policy:**

Affirmed.

**Final Justification:**

The author has addressed my concerns, so I will maintain a positive evaluation. Based on the author’s responses to other reviewers, if the details mentioned in those responses are further refined in the final version, I believe the paper meets the acceptance standards of ICML. Therefore, I will raise my score to 5.

**Key Questions For Authors:**

• High-Dimensional Scaling: You noted that CMD-P increases as the target proteome dimensionality expands, suggesting difficulty in capturing large-scale inter-protein regulatory networks. Have you conducted preliminary experiments incorporating explicit biological priors (such as Protein-Protein Interaction graphs) into the Stage 1 encoder to mitigate this?

• Two-Stage vs. Joint Training: Stage 1 (Manifold Rectification) and Stage 2 (Flow Matching) are trained sequentially. Can you elaborate on why end-to-end joint training was avoided? Would backpropagating the CFM loss directly into the VAE encoder disrupt the noise-absorbing properties of the ZINB decoder?

• Sensitivity to Consistency Masking: The consistency loss $L_{cons}$ involves randomly masking a ratio $r$ of expressed genes up to $r_{max}$. Given that some datasets like GSE100866 already exhibit extreme baseline RNA sparsity (e.g., 96%), how sensitive is the generative fidelity to the hyperparameter $r_{max}$? Does aggressive masking in already sparse regimes risk destroying the semantic condition embedding entirely?

**Limitations:**

yes

**Strengths And Weaknesses:**

Soundness

•	Strengths: The theoretical foundation is exceptionally rigorous. The authors formally prove how fitting deterministic models to noisy data mathematically necessitates high local sensitivity, leading to geometric roughness. Furthermore, the derivation of how the ZINB decoder provides "Variance-Adaptive Gradient Rescaling" to smoothly bound the Lipschitz constant is elegant and convincing. The empirical validation is also comprehensive, covering zero-shot generalization across cell types, batches, and time points.

•	Weaknesses: The authors honestly acknowledge a scaling limitation: as the target proteome dimensionality increases, preserving protein-level structural fidelity (measured by CMD-P) becomes challenging. Relying solely on the implicit structural guidance of the current architecture may not suffice for massive, whole-proteome translations without explicit graph-based priors.

Presentation

•	Strengths: The manuscript is superbly organized and a pleasure to read. The conceptual contrast between the "Original Manifold" and the "Rectified Manifold" is intuitively explained. The ablation studies are logically structured to isolate the specific contributions of the probabilistic formulation, the KL divergence, and the consistency loss. Additionally, the Kernel Density Estimation (KDE) visualizations convincingly illustrate the distributional fidelity.

•	Weaknesses: The methodology relies on a two-stage decoupled training pipeline (VAE first, then CFM). The manuscript lacks a brief discussion on the optimization dynamics and potential failure modes if one were to attempt end-to-end joint training.

Significance

•	Strengths: Inferring protein abundance from highly accessible RNA data is a critical computational necessity due to the high costs and throughput limitations of current proteomics . scChord provides immense biological value by faithfully recovering true biological heterogeneity—such as the bimodal distribution of CD4 and the heavy-tailed continuous spectrum of CD45RA—which deterministic regression models over-smooth and other generative models distort with zero-inflation artifacts.

Originality

•	Strengths: Identifying the geometric incompatibility between deterministic mapping and single-cell stochasticity is a profound insight. Using a probabilistic decoder merely as a "manifold rectifier" to precondition the latent space for subsequent continuous generative dynamics is a highly creative synthesis of VAEs and Flow Matching. It addresses the root cause of ODE solver stiffness in biological applications.

---

> ### Author Rebuttal · Authors · 2026-03-31
>
> We sincerely thank you for your exceptionally detailed and insightful review. We are deeply encouraged by your strong endorsement of our theoretical foundation, particularly your formalization of geometric roughness and the variance-adaptive gradient rescaling mechanism. Your precise understanding of the ZINB decoder as a geometric preconditioner perfectly aligns with the core contribution of scChord. Below, we provide detailed responses to your excellent questions regarding scalability, optimization dynamics, and robustness.
>
> **1. High-Dimensional Scaling and Explicit Biological Priors**
>
> > _Have you conducted preliminary experiments incorporating explicit biological priors (such as PPI graphs) into the Stage 1 encoder to mitigate this?_
>
> We are thrilled to receive this highly perceptive suggestion, as it coincides precisely with our recent and ongoing research efforts. As our empirical analysis indicates, as the dimensionality of the target proteome expands, relying solely on implicit geometric constraints becomes insufficient to capture the vast and complex inter-protein regulatory networks.
>
> We extracted Protein-Protein Interaction (PPI) data from the STRING database and attempted to model the target proteins using Graph Attention Networks (GAT) in the Stage 1 encoder, replacing the original MLP backbone. Encouragingly, injecting this explicit biological prior indeed had a positive impact on prediction performance, partially mitigating the degradation of structural fidelity (CMD-P) in high-dimensional settings. However, compared to our current lightweight version, the GAT backbone introduces significantly larger training and inference times along with a heavier memory burden. We will continue to focus on this issue to find a more computationally reasonable solution, plan to upload this exploratory code named **gat_fm** to our same anonymous repository (https://anonymous.4open.science/r/scChord-A288/) in the near future, and will fully open-source it after the review period ends. Prompted by your valuable feedback, we have thoroughly discussed this graph-based scaling strategy in the "Limitations and Future Work" section of the revised manuscript.
>
> **2. Two-Stage vs. Joint Training**
>
> > _Can you elaborate on why end-to-end joint training was avoided? Would backpropagating the CFM loss directly into the VAE encoder disrupt the noise-absorbing properties of the ZINB decoder?_
>
> Your intuition regarding the optimization dynamics is remarkably accurate. In our ablation study (Variants A and B in Table 3), we implicitly attempted two end-to-end solutions, both resulting in significant performance degradation. As you astutely hypothesized, fully joint training introduces two severe failure modes:
>
> - **Destruction of Noise-Absorbing Properties:** The strong regression gradients from the Flow Matching objective would backpropagate directly into the VAE encoder, bypassing the ZINB decoder's safeguards and forcing the model to re-encode measurement noise into $z$ to satisfy point-wise matching—directly causing the "uncertainty collapse" we set out to resolve.
>
> - **Moving Target and Optimization Instability:** Joint training causes the target manifold to continuously shift as the VAE updates, severely destabilizing the flow dynamics and inducing high-curvature ODE trajectories with severe solver stiffness.
>
>
> Following your excellent suggestion, we have explicitly supplemented the revised manuscript with a thorough discussion of these optimization dynamics and failure modes.
>
> **3. Sensitivity to Consistency Masking in Sparse Regimes**
>
> > _Considering that some datasets (like GSE100866) already exhibit extreme baseline RNA sparsity, does aggressive masking risk destroying the semantic condition embedding entirely?_
>
> This is a highly intuitive and valid concern that we greatly appreciate. Counterintuitively, however, aggressive masking is precisely the key to _preventing_ the semantic embedding from collapsing under extreme sparsity regimes. In datasets like GSE100866 (where baseline RNA sparsity exceeds 96%), the few non-zero readouts are heavily corrupted by technical dropouts; without masking, the encoder tends to overfit to the exact positions of these unreliable non-zero entries. By randomly masking a proportion of already-expressed genes, we force the encoder to rely on macro-level co-expression modules rather than individual gene readouts. Ablation Variant G empirically validates this: removing the consistency loss on GSE100866 results in a significant increase in RMSE and deterioration of CMD. We have added a clarifying explanation in the manuscript to highlight why this mechanism is particularly critical for highly zero-inflated data.

---

> > ### Author Rebuttal · Reviewer_Fv96 · 2026-04-01
> >
> > The author has addressed my concerns, so I will retain a positive evaluation. I hope the authors will prepare strong rebuttals to the other reviewers.
> >
> > ----
> >
> > Based on the author’s responses to the other reviewers, if all the details and experiments mentioned in the rebuttal can be properly incorporated into the final version of the paper, I believe the work meets the acceptance standard of ICML. Therefore, I will raise my score to 5.

---

> > > ### Author Response · Authors · 2026-04-02
> > >
> > > We are truly grateful for your warm and encouraging reply. Your supportive words gave us the exact motivation we needed, and following your advice, we have put our utmost effort into preparing detailed and strong rebuttals for the other reviewers.
> > >
> > > In particular, we want to express our special thanks for your insightful suggestion regarding the attempt to incorporate GAT in stage 1. We were absolutely thrilled to receive such a brilliant piece of advice, as it profoundly inspires us and provides clear, significant guidance for our future improvements.
> > >
> > > We want to sincerely say that, whatever the final result of this conference may be, we deeply appreciate the dedication you and the other reviewers have shown. Your constructive feedback has been invaluable in making this a much better piece of work.
> > >
> > > Should you have any lingering thoughts or new questions, we would love the opportunity to discuss them further with you. Thank you again for your time, your kindness, and your expertise.
> > >
> > > Best regards,
> > >
> > > _Submission 22756 Authors_

---

### Official Review · Reviewer_LEjJ · 2026-03-13

**Soundness:** 3
**Presentation:** 3
**Significance:** 3
**Originality:** 3
**Overall Recommendation:** 4
**Confidence:** 4

**Summary:**

This paper proposes scChord to infer protein abundance based on RNA data. The model is based on a conditional flow matching method motivated by probabilistic manifold learning, which disentangle technical noise and true biological signals.

**Compliance With Llm Reviewing Policy:**

Affirmed.

**Final Justification:**

The rebuttal addressed my concerns so I raised by score from 3 to 4.

**Key Questions For Authors:**

Please check weakness part.

**Limitations:**

High dimensionality is discussed as a limitation. The authors did give some alternative solutions like incorporating pathway level priors of hierarchical structure.

**Strengths And Weaknesses:**

**Strength**

The biological problem studied in this paper is relevant, of great practical value to the field.

The proposed method is well motivated, from observations to hypothesis then to model design.

**Weakness**

The paper assumes the RNA data are "readily available". However, in practice, RNA data can also be messy. Can the authors discuss when RNA data are in low quality or noisy, how does it affect the proposed method for proteins?

The computational complexity or actual runtime is not discussed. I suppose the proposed approach is slower than deterministic models.

The authors argue that the method is to solve a stochastic inverse problem. However, existing empirical results do not include rare cell types or tissues with low RNA-protein correlation, which makes the inverse problem more severe.

Based on the experiments for batch effect, can the authors go one step further: does the method generalize to unseen batches?

 The experiments are more focused on inference accuracy from the global point of view, i.e., protein distribution. However, for certain downstream tasks, precise per-cell accuracy is more interesting and deserves some discussion.

There are a lot of missing spaces, especially those before ( in citations.

---

> ### Author Rebuttal · Authors · 2026-03-31
>
> We are genuinely grateful for the reviewer's thoughtful and constructive feedback. Your questions address key practical challenges in deploying generative models in computational biology, and we address each point below.
>
> **1. Noisy/low-quality RNA data.**
> We completely agree this is a critical real-world concern that we should address more explicitly. Our Stage 2 training incorporates Transcriptomic Consistency Regularization ($\mathcal{L}_{cons}$), which explicitly simulates technical dropouts by randomly masking expressed genes and forcing the model to align corrupted embeddings to full profiles. Ablation results (Table 3, Variant G) confirm that removing this constraint significantly increases RMSE and worsens CMD, demonstrating scChord learns robust biological semantics rather than overfitting to technical noise. We have expanded this discussion in the revision.
>
> **2. Computational complexity and runtime.**
> We thank the reviewer for raising this practical concern. We benchmarked all models at scales of 1k, 10k, 50k, and 100k cells (GSE164378-3P, NVIDIA RTX 4080); full scaling curves are in the revised version and in our anonymous repository linked at the end of the paper.
>
> | Method | Train (s) | CPU Mem (MB) | GPU Mem (MB) | Infer (s/200) |
> |---|---|---|---|---|
> | cTp_net | — | — | — | 0.259 |
> | sciPENN | — | — | — | 0.002 |
> | Dengkw | 195 | 4,910 | 0 | 0.023 |
> | Liger | 403 | 36,356 | 0 | 0.026 |
> | scArches | 1,419 | 23,779 | 833 | 0.069 |
> | scVAEIT | 2,923 | 9,968 | 712 | 0.465 |
> | totalVI | 1,297 | 22,140 | 392 | 0.030 |
> | scChord | 1,636 | 10,766 | 229 | 0.488 |
> *(scChord total = Stage 1: 576s + Stage 2: 1,061s.)*
> **Note:** For cTp_net and sciPENN, training and memory metrics are unavailable because both methods crashed with out-of-memory errors (exceeding 64GB RAM) at the 100k cell scale.
>
> We would like to offer an honest and balanced assessment: (1) Training time: We readily acknowledge that non-deep-learning methods (Dengkw, Liger) train substantially faster, which is an expected and fair trade-off for their lower modeling capacity. Among deep generative models, however, scChord (~1,636s) is competitive—1.8× faster than scVAEIT and broadly comparable to totalVI. (2) Inference speed: We honestly acknowledge that scChord's inference latency is the highest among all baselines, and this is a genuine limitation stemming from the multi-step ODE integration required by the Flow Matching paradigm. This overhead is further linked to our default high-precision solver (dopri5). We are actively working on two complementary mitigation strategies: reducing the number of integration steps, and replacing dopri5 with a more efficient fixed-step solver (e.g., Euler or midpoint), both of which yield substantial speedups with only marginal accuracy degradation in our preliminary tests. (3) Memory efficiency: scChord's most notable advantage lies in its hardware footprint. Under a uniform batch size of 512, it achieves the lowest GPU peak memory (229 MB). Its CPU memory scales linearly to ~10.7 GB at 100k cells, far below Liger (36 GB) and totalVI (22 GB), making it accessible without specialized infrastructure.
>
> **3. Rare cell types and low RNA-protein correlation.**
> We greatly appreciate this insightful observation. Our CD45RA evaluation directly speaks to this: its heavy-tailed distribution represents low-frequency memory T-cell states, and Figure 7(b) shows that deterministic baselines suffer mode collapse on these rare populations while scChord faithfully recovers the full spectrum. We acknowledge, however, that a dedicated evaluation on tissues with systematically low RNA-protein correlation remains absent from our current experiments, and we agree this is an important limitation and future direction. We have made both the rare-cell-type connection and this limitation explicit in the revised text.
>
> **4. Generalization to unseen batches.**
> We are glad the reviewer raised this critical question. In Section 4.1 we performed a strict leave-one-group-out evaluation on GSE164378-5P with a cross-donor hold-out axis, meaning the model is tested on a completely unseen donor batch. Figure 4 shows scChord maintains strong performance in this zero-shot setting, suggesting it learns batch-invariant biological representations. We have made this aspect considerably more prominent in the revision.
>
> **5. Per-cell accuracy for downstream tasks.**
> We fully agree and thank the reviewer for this important emphasis. Tables 1 and 2 already report Cell-wise Pearson Correlation (PCC-C) and Cell-wise CMD (CMD-C); scChord achieves sota per-cell accuracy. We have added a dedicated discussion on downstream task implications—including cell clustering and differential abundance analysis—to the revised manuscript.
>
> **6. Missing spaces before citations.**
> We sincerely apologize for this careless oversight, caused by a LaTeX citation macro issue. We have thoroughly proofread the revision and corrected all instances.

---

> > ### Author Rebuttal · Reviewer_LEjJ · 2026-04-03
> >
> > Thank the authors to respond to my questions carefully. All my concerns were addressed and I have raised my score. I strongly recommend the authors to include the runtime discussion in the next version.

---

> > > ### Author Response · Authors · 2026-04-03
> > >
> > > We sincerely thank the reviewer for the careful and open-minded evaluation, and for raising the score. We will make sure to include a thorough runtime and computational efficiency discussion in the revised manuscript, as suggested. We truly appreciate the constructive feedback throughout this process!
> > >
> > > Best regards,
> > >
> > > _Submission 22756 Authors_

---

### Decision · Program_Chairs · 2026-04-30

**Decision:**

Accept (regular)

**Comment:**

This paper addresses a fundamental issue with current approaches to RNA-to-protein inference. At a high level, the issue stems from the combination of enforcing deterministic constraints with noisy input data, which leads to unstable learning dynamics. The paper proposes a sophisticated and biologically well-motivated modeling approach that cleanly separates biological signals from technical noise and overdispersion in the raw data. After the discussion period, the reviewers were impressed with a number of aspects of the proposed approach, including its biological motivation, theoretical sophistication, and computational efficiency, and were moreover largely convinced by the paper's empirical work.